

# A survey of the state of the practice for research software in the United States

Jeffrey C. Carver[1], Nic Weber[2], Karthik Ram[3], Sandra Gesing[4] and Daniel S. Katz[5]

[1] Computer Science, University of Alabama, Tuscaloosa, AL, United States of America
[2] Information School, University of Washington, Seattle, WA, United States of America
[3] Berkeley Institute for Data Science, University of California, Berkeley, Berkeley, CA, United States of America
[4] Discovery Partners Institute, Chicago, IL, United States of America
[5] NCSA & CS & ECE & iSchool, University of Illinois at Urbana-Champaign, Urbana, IL, United States of America

## ABSTRACT

Research software is a critical component of contemporary scholarship. Yet, most research software is developed and managed in ways that are at odds with its long-term sustainability. This paper presents findings from a survey of 1,149 researchers, primarily from the United States, about sustainability challenges they face in developing and using research software. Some of our key findings include a repeated need for more opportunities and time for developers of research software to receive training. These training needs cross the software lifecycle and various types of tools. We also identified the recurring need for better models of funding research software and for providing credit to those who develop the software so they can advance in their careers. The results of this survey will help inform future infrastructure and service support for software developers and users, as well as national research policy aimed at increasing the sustainability of research software.

# INTRODUCTION

In almost all areas of research, from hard sciences to the humanities, the processes of collecting, storing, and analyzing data and of building and testing models have become increasingly complex. Our ability to navigate such complexity is only possible because of the existence of specialized software, often referred to as *research software*. Research software plays such a critical role in day to day research that a comprehensive survey reports 90–95% of researchers in the US and the UK rely upon it and more than 60% were unable to continue working if such software stopped functioning (*Hettrick, 2014*). While the research community widely acknowledges the importance of research software, the creation, development, and maintenance of research software is still *ad hoc* and improvised, making such infrastructure fragile and vulnerable to failure.

In many fields, research software is developed by academics who have varying levels of training, ability, and access to expertise, resulting in a highly variable software landscape. As researchers are under immense pressure to maintain expertise in their research domains, they have little time to stay current with the latest software engineering practices. In

Corresponding author
Jeffrey C. Carver, carver@cs.ua.edu

addition, the lack of clear career incentives for building and maintaining high quality software has made research software development unsustainable. The lack of career incentives has occurred partially because the academic environment and culture have developed over hundreds of years, while software has only recently become important, in some fields over the last 60+ years, but in many others, just in the last 20 or fewer years (*Foster, 2006*).

Further, only recently have groups undertaken efforts to promote the role of research software (*e.g.*, the Society of Research Software Engineers (https://society-rse.org), the US Research Software Engineer Association (https://us-rse.org)) and to train researchers in modern development practices (*e.g.*, the Carpentries (https://carpentries.org), IRIS-HEP (https://iris-hep.org), and MolSSI (https://molssi.org)). While much of the development of research software occurs in academia, important development also occurs in national laboratories and industry. Wherever the development and maintenance of research software occurs, that software might be released as open source (most likely in academia and national laboratories) or it might be commercial/closed source (most likely in industry, although industry also produces and contributes to open source).

The open source movement has created a tremendous variety of software, including software used for research and software produced in academia. It is difficult for researchers to find and use these solutions without additional work (*Joppa et al., 2013*). The lack of standards and platforms for categorizing software for communities often leads to re-developing instead of reusing solutions (*Howison et al., 2015*). There are three primary classes of concerns, pervasive across the research software landscape, that have stymied this software from achieving maximum impact.

- *Functioning of the individual and team*: issues such as training and education, ensuring appropriate credit for software development, enabling publication pathways for research software, fostering satisfactory and rewarding career paths for people who develop and maintain software, and increasing the participation of underrepresented groups in software engineering.
- *Functioning of the research software*: supporting sustainability of the software; growing community, evolving governance, and developing relationships between organizations, both academic and industrial; fostering both testing and reproducibility, supporting new models and developments (*e.g.*, agile web frameworks, Software-as-a-Service), supporting contributions of transient contributors (*e.g.*, students), creating and sustaining pipelines of diverse developers.
- *Functioning of the research field itself*: growing communities around research software and disparate user requirements, cataloging extant and necessary software, disseminating new developments and training researchers in the usage of software.

In response to some of the challenges highlighted above, the US Research Software Sustainability Institute (URSSI) (http://urssi.us) conceptualization project, funded by NSF, is designing an institute that will help with the problem of sustaining research software. The overall goal of the conceptualization process is *to bring the research software community together to determine how to address known challenges to the development and sustainability*

*of research software and to identify new challenges that need to be addressed.* One important starting point for this work is to understand and describe the current state of the practice in the United States relative to those important concerns. Therefore, in this paper we describe the results of a community survey focused on this goal.

## BACKGROUND

Previous studies of research software have often focused on the development of cyberinfrastructure (*Borgman, Wallis & Mayernik, 2012*) and the various ways software production shapes research collaboration (*Howison & Herbsleb, 2011*; *Howison & Herbsleb, 2013*; *Paine & Lee, 2017*). While these studies provide rich contextual observations about research software development processes and practices, their results are difficult to generalize because they often focus either on small groups or on laboratory settings. Therefore, there is a need to gain a broader understanding of the research software landscape in terms of challenges that face individuals seeking to sustain research software.

A number of previous surveys have provided valuable insight into research software development and use, as briefly described in the next subsection. Based on the results of these surveys and from other related literature, the remainder of this section motivates a series of research questions focused on important themes related to the development of research software. The specific questions are based on the authors' experience in common topics mentioned in the first URSSI workshop (*Ram et al., 2018*) as well as previously published studies of topics of interest to the community (*Katz et al., 2019*; *Fritzsch, 2019*).

### Previous surveys

The following list provides an overview of the previous surveys on research software, including the context of each survey. Table 1 summarizes the surveys.

- *How do Scientists Develop and Use Scientific Software?* (*Hannay et al., 2009*) describes the results of a survey of 1972 scientists who develop and use software. The survey focused on questions about (1) how and when scientists learned about software development/use, (2) the importance of developing/using software, (3) time spent developing/using software, (4) hardware platforms, (5) user communities, and (6) software engineering practices.
- *How Do Scientists Develop Scientific Software? An External Replication* (*Pinto, Wiese & Dias, 2018*) is a replication of the previous study (*Hannay et al., 2009*) conducted ten years later. The replication focused on scientists who develop R packages. The survey attracted 1,553 responses. The survey asked very similar questions to the original survey, with one exception. In addition to replicating the original study, the authors also asked respondents to identify the "most pressing problems, challenges, issues, irritations, or other 'pain points' you encounter when developing scientific software." A second paper, *Naming the Pain in Developing Scientific Software* (*Wiese, Polato & Pinto, 2020*), describes the results of this question in the form of a taxonomy of 2,110 problems that are either (1) technical-related, (2) social-related, or (3) scientific-related.
- *A Survey of Scientific Software Development* (*Nguyen-Hoan, Flint & Sankaranarayana, 2010*) surveyed researchers in Australia working in multiple scientific domains. The

**Table 1 Previous surveys.**

| Study | Focus | Respondents |
|---|---|---|
| *Hannay et al. (2009)* | How scientists develop and use software | 1972 |
| *Pinto, Wiese & Dias (2018)* | Replication of *Hannay et al. (2009)* | 1553 |
| *Wiese, Polato & Pinto (2020)* | Additional results from *Pinto, Wiese & Dias (2018)* focused on problems encountered when developing scientific software | 1577 |
| *Nguyen-Hoan, Flint & Sankaranarayana (2010)* | Software development practices of scientists in Australia | 60 |
| *Prabhu et al. (2011)* | Practice of computational science in one large university | 114 |
| *Joppa et al. (2013)* | Researchers in species domain modeling with varying levels of expertise | ~450 |
| *Carver et al. (2013)* | Software engineering knowledge and training among computational scientists and engineers | 141 |
| *Hettrick (2018)*; *Hettrick (2014)* | Use of software in Russell Group Universities in the UK | 417 |
| *Jay, Sanyour & Haines (2016)* | How scientists publish code | 65 |
| *Nangia & Katz (2017)* | Use of software and software development training in US Postdoctoral Association | 209 |
| *AlNoamany & Borghi (2018)* | How the way researchers use, develop, and share software impacts reproduciblity | 215 |
| *Philippe et al. (2019)* | Research Software Engineers | ~1000 |

survey focused on programming language use, software development tools, development teams and user bases, documentation, testing and verification, and non-functional requirements.

- *A Survey of the Practice of Computational Science* (*Prabhu et al., 2011*) reports the results of interviews of 114 respondents from a diverse set of domains all working at Princeton University. The interviews focused on three themes: (1) programming practices, (2) computational time and resource usage, and (3) performance enhancing methods.

- *Troubling Trends in Scientific Software* (*Joppa et al., 2013*) reports on the results from about 450 responses working in a specific domain, species distribution modeling, that range from people who find software difficult to use to people who are very experienced and technical. The survey focused on understanding why respondents chose the particular software they used and what other software they would like to learn how to use.

- *Self-Perceptions About Software Engineering: A Survey of Scientists and Engineers* (*Carver et al., 2013*) reports the results from 141 members of the Computational Science & Engineering community. The primary focus of the survey was to gain insight into whether the respondents thought they knew enough software engineering to produce high-credibility software. The survey also gathered information about software engineering training and about knowledge of specific software engineering practices.

- *"Not everyone can use Git:" Research Software Engineers' recommendations for scientist-centered software support (and what researchers really think of them)* *Jay, Sanyour & Haines (2016)* describes a study that includes both Research Software Engineers and domain researchers to understand how scientists publish code. The researchers began

by interviewing domain scientists who were trying to publish their code to identify the barriers they faced in publishing their code. Then they interviewed Research Software Engineers to understand how they would address those barriers. Finally, they synthesized the results from the Research Software Engineer interviews into a series of survey questions sent to a larger group of domain researchers.

- *It's impossible to conduct research without software, say 7 out of 10 UK researchers* (*Hettrick, 2018*; *Hettrick, 2014*) describes the results of 417 responses to a survey of 15 Russel Group Universities in the UK. The survey focused on describing the characteristics of software use and software development within research domains. The goal was to provide evidence regarding the prevalence of software and its fundamental importance for research.

- *Surveying the US National Postdoctoral Association Regarding Software Use and Training in Research* (*Nangia & Katz, 2017*) reports on the results of 209 responses to provide insight into the role of software in conducting research at US universities. The survey focused on the respondents' use of research software and the training they have received in software development.

- *Towards Computational Reproducibility: Researcher Perspectives on the Use and Sharing of Software* (*AlNoamany & Borghi, 2018*) reports on the results from 215 respondents across a range of disciplines. The goal of the survey was to understand how researchers create, use, and share software. The survey also sought to understand how the software development practices aligned with the goal of reproducibility.

- *SSI International RSE Survey* (*Philippe et al., 2019*) reports on the results from approximately 1000 responses to a survey of research software engineers from around the world. The goal of the survey is to describe the current state of research software engineers related to various factors including employment, job satisfaction, development practices, use of tools, and citation practices.

## Software engineering practices

Based on the results of the surveys described in the previous subsection, we can make some observations about the use of various software engineering practices employed while developing software. The set of practices research developers find useful appear to have some overlap and some difference from those practices employed by developers of business or IT software. Interestingly, the results of the previous surveys do not paint a consistent picture regarding the importance and/or usefulness of various practices. Our current survey is motivated by the inconsistencies in previous results and the fact that some key areas are not adequately covered by previous surveys. Here we highlight some of the key results from these previous surveys, organized roughly in the order of the software engineering lifecycle.

### *Requirements*

The findings of two surveys (*Pinto, Wiese & Dias, 2018*; *Hannay et al., 2009*) reported both that requirements were important to the development of research software but also that they were one of the least understood phases. Other surveys reported that (1) requirements management is the most difficult technical problem (*Wiese, Polato & Pinto,*

*2020*) and (2) the amount of requirements documentation is low (*Nguyen-Hoan, Flint & Sankaranarayana, 2010*).

### Design

Similar to requirements, surveys reported that design was one of the most important phases (*Hannay et al., 2009*) and one of the least understood phases (*Pinto, Wiese & Dias, 2018*; *Hannay et al., 2009*). In addition, other surveys reported that (1) testing and debugging are the second most difficult technical problem (*Nguyen-Hoan, Flint & Sankaranarayana, 2010*) and (2) the amount of design documentation is low (*Wiese, Polato & Pinto, 2020*).

### Testing

There were strikingly different results related to testing. A prior survey of research software engineers found almost 2/3 of developers do their own testing, but less than 10% reported the use of formal testing approaches (*Philippe et al., 2019*). Some surveys (*Pinto, Wiese & Dias, 2018*; *Hannay et al., 2009*) reported that testing was important. However, another survey reported that scientists do not regularly test their code (*Prabhu et al., 2011*). Somewhere in the middle, another survey reports that testing is commonly used, but the use of integration testing is low (*Nguyen-Hoan, Flint & Sankaranarayana, 2010*).

*Software engineering practices summary.* This discussion all leads to the first research question: *RQ1: What activities do research software developers spend their time on, and how does this impact the perceived quality and long-term accessibility of research software?*

## Software tools and support

Development and maintenance of research software includes both the use of standard software engineering tools such as version control (*Milliken, Nguyen & Steeves, 2021*) and continuous integration (*Shahin, Babar & Zhu, 2017*). In addition, these tasks require custom libraries developed for specific analytic tasks or even language-specific interpreters that ease program execution.

Previous surveys have asked researchers and research software engineers about the most frequently used open-source software in development. Surveys of research software developers and users have reported the use of standard software languages and even the types of tools used in analaysis (*AlNoamany & Borghi, 2018*), but there has been relatively little description of the tools upon which research software developers depend, and to what extent these tools are seen by developers as supporting sustainable research software practices. We therefore seek to understand tool usage and support in a second research question that asks: *RQ2: What tools do research software developers use and what additional tools are needed to support sustainable development practices?*

## Education and training

While researchers often develop research software for the express purpose of conducting research, previous studies demonstrate that these researchers are rarely purposely trained to develop software. A 2012 survey reported that research software developers had little formal training and were mostly self-taught (*Carver et al., 2013*).

A UK survey (*Hettrick, 2018*; *Hettrick, 2014*) reported only 55% of respondents had some software development training. Of those only 40% had formal training, with 15% being self-taught. In addition, only 2% of respondents who develop their own software had no training in software development. The 2017 survey of US National Postdoctoral Association (*Nangia & Katz, 2017*) found similar results: while 95% of the respondents used research software, 54% reported they had not received any training in software development. When analyzed by gender (self reported binary of men and women) these two surveys show remarkable similarities in the gap of training for men (63% in the UK and 63% in the US) and women (39% in the UK and 32% in US). The *AlNoamany & Borghi (2018)* survey reported similar results related to training: 53% of respondents had formal training in coding conventions and best practices. The *Hannay et al. (2009)* survey along with the *Pinto, Wiese & Dias (2018)* replication reported slightly less positive results. Regarding different mechanisms for learning about software development, 97% and 99% of the respondents thought *self-study* was important or very important, while only 13% and 22% found *formal training* to be important or very important.

The results of these prior surveys suggest that research software developers are either unaware of their need for or may not have access to sufficient formal training in software development. In addition, the results of the *Joppa et al. (2013)* survey indicate that most respondents want increased computational skills. The authors advocate for formal training in software engineering as part of the University science curriculum.

Therefore, we pose the following research question that guides our specific survey questions related to training –*RQ3: What training is available to research software developers and does this training meet their needs?*

## Funding and institutional support

One of the key sustainability dilemmas for research software is the lack of direct financial support for development and maintenance. Successful research grants often focus on the merits of a new idea and the potential novel scientific or scholarly contribution of progress made on that idea. However, both institutions that support research (*e.g.*, universities and national laboratories) and grant-making bodies that fund research (*e.g.*, federal agencies and philanthropic organizations) often fail to recognize the central importance of software development and maintenance in conducting novel research (*Goble, 2014*). In turn, there is a little direct financial support for the development of new software or the sustainability of existing software upon which research depends (*Katerbow et al., 2018*). In particular, funding agencies typically have not supported the continuing work needed to maintain software after its initial development. This lack of support is despite increasing recognition of reproducibility and replication crises that depend, in part, upon reliable access to the software used to produce a new finding (*Hocquet & Wieber, 2021*).

In reaction to a recognized gap in research funding for sustainable software, many projects have attempted to demonstrate the value of their work through traditional citation and impact analysis (*Anzt et al., 2021*) as well as through economic studies. An example of the latter was performed by a development team of the widely used AstroPy packages in Astronomy. Using David A. Wheeler's *SLOCCount* method for economic impact of

open-source software they estimate the cost of reproducing AstroPy to be approximately $8.5 million and the annual economic impact on astronomy alone to be approximately $1.5 million (*Muna et al., 2016*).

There is, recently, increased attention from funders on the importance of software maintenance and archiving, including the Software Infrastructure for Sustained Innovation (SI2) program at NSF, the NIH Data Commons (which includes software used in biomedical research), the Alfred P. Sloan Foundation's Better Software for Science program, and the Chan Zuckerberg Initiative's Essential Open Source Software for Science program which provide monetary support for the production, maintenance, and adoption of research software. Despite encouraging progress there is still relatively little research that focuses specifically on how the lack of direct financial support for software sustainability impacts research software engineers and research software users. We seek to better understand this relationship through two specific research questions that focus on the impact of funding on software sustainability: *RQ4a: What is the available institutional support for research software development?* and *RQ4b: What sources of institutional funding are available to research software developers?*

## Career paths

While most of previous surveys did not address the topic of career paths, the survey of research software engineers (*Philippe et al., 2019*) did briefly address this question. Because the results differ across the world and our paper focuses on the US, we only report results for respondents in the US. First, 57% of respondents were funded by grants and 47% by institutional support. Second, respondents had been in their current position for an average of 8.5 years. Last, 97% were employed full-time.

Because of the lack of information from prior surveys, we focus the rest of this discussion on other work to provide background. In 2012, the Software Sustainability Institute (SSI) organized the Collaborations Workshop (http://software.ac.uk/cw12) that addressed the question: *why is there no career for software developers in academia?* The work of the participants and of the SSI's policy team led to the foundation of the UK RSE association and later to the Society of Research Software Engineering. RSEs around the world are increasingly forming national RSE associations, including the US Research Software Engineer Association (US-RSE) (http://us-rse.org/).

Current evaluation and promotion processes in academia and national labs typically follow the traditional pattern of rewarding activities that include publications, funding, and advising students. However, there are other factors that some have considered. Managers of RSE teams state that when hiring research developers, it is important that those developers are enthusiastic about research topics and have problem-solving capabilities (https://cosden.github.io/improving-your-RSE-application). Another factor, experience in research software engineering, can be evaluated by contributions to software in platforms like GitHub. However, while lines of code produced, number of solved bugs, and work hours may not be ideal measures for developer productivity, they can provide insight into the sustainability and impact of research software, *i.e.,* the presence of an active community behind a software package that resolves bugs and interacts with users is part of sustainability

of software and impact on research (https://github.com/Collegeville/CW20/blob/master/WorkshopResources/WhitePapers/gesing-team-organization.pdf). In addition, CaRCC (the Campus Research Computing Consortium) has defined job families and templates for job positions that can be helpful both for hiring managers and HR departments that want to recognize the role of RSEs and HPC Facilitators in their organizations (https://carcc.org/wp-content/uploads/2019/01/CI-Professionalization-Job-Families-and-Career-Guide.pdf).

However, there is still not a clearly defined and widely accepted career path for research software engineers in the US. We pose the following research question that guides our specific survey questions related to career paths –*RQ5: What factors impact career advancement and hiring in research software?*

## Credit

While most of the previous surveys did not address the topic of credit, the survey of research software engineers (*Philippe et al., 2019*) does contain a question about how researchers are acknowledged when their software contributes to a paper. The results showed that 47% were included as a co-author, 18% received only an acknowledgement, and 21% received no mention at all. Because of the lack survey results related to credit, we focus on other work to provide the necessary background.

The study of credit leads to a set of interlinked research questions. We can answer these questions by directly asking software developers and software project collaborators to provide their insights. Here we take a white box approach and examine the inside of the box.

- How do individuals want their contributions to software projects to be recognized, both as individuals and as members of teams?
- How do software projects want to record and make available credit for the contributions to the projects?

In addition, these respondents can help answer additional questions from their perspective as someone external to other organizations. Here we can only take a black box approach and examine the box from the outside. (A white box approach would require a survey of different participants.)

- How does the existing ecosystem, based largely on the historical practices related to contributions to journal and conference papers and monographs, measure, store, and disseminate information about contributions to software?
- How does the existing ecosystem miss information about software contributions?
- How do institutions (*e.g.*, hiring organizations, funding organizations, professional societies) use the existing information about contributions to software, and what information is being missed?

We also recognize that there are not going to be simple answers to these questions (*CASBS Group on Best Practices in Science, 2018*; *Albert & Wager, 2009*), and that any answers will likely differ to some extent between disciplines (*Dance, 2012*). Many professional societies and publishers have specific criteria for authorship of papers (*e.g.*, they have

made substantial intellectual contributions, they have participated in drafting and/or revision of the manuscript, they agree to be held accountable for any issues relating to correctness or integrity of the work (*Association for Computing Machinery, 2018*), typically suggesting that those who have contributed but do not meet these criteria be recognized *via* an acknowledgment. While this approach is possible in a paper, there is no equivalent for software, other than papers about software. In some disciplines, such as those where monographs are typical products, there may be no formal guidelines. Author ordering is another challenge. The ordering of author names typically has some meaning, though the meaning varies between disciplines. Two common practices are alphabetic ordering, such as is common in economics (*Weber, 2018*) and ordering by contribution with the first author being the main contributor and the last author being the senior project leader, as occurs in many fields (*Riesenberg & Lundberg, 1990*). The fact that the contributions of each author is unclear has led to activities and ideas to record their contributions in more detail (*Allen et al., 2014*; *The OBO Foundry, 2020*; *Katz, 2014*).

Software in general has not been well-cited (*Howison & Bullard, 2016*), in part because the scholarly culture has not treated software as something that should be cited, or in some cases, even mentioned. The recently-perceived reproducibility crisis (*Baker, 2016*) has led to changes, first for data (which also was not being cited (*Task Group on Data Citation Standards and Practices, 2013*)) and more recently for software. For software, these changes include the publication of software papers , both in general journals and in journals that specialize in software papers (*e.g.*, the Journal of Open Source Software *Smith et al., 2018*), as well as calls for direct software citation (*Smith et al., 2016*) along with guidance for those citations (*Katz et al., 2021*). Software, as a digital object, also has the advantage that it is usually stored as a collection of files, often in a software repository. This fact means that it is relatively simple to add an additional file that contains metadata about the software, including creators and contributions, in one of a number of potential styles (*Wilson, 2013*; *Druskat, 2020*; *Jones et al., 2017*). This effort has recently been reinforced by GitHub, who have made it easy to add such metadata to repositories and to generate citations for those repositories (*Smith, 2021*).

Therefore, we pose the following research questions to guide our specific survey questions related to credit – *RQ6a What do research software projects require for crediting or attributing software use?* and *RQ6b: How are individuals and groups given institutional credit for developing research software?*

## Diversity

Previous research has found that both gender diversity and tenure (length of commitment to a project) are positive and significant predictors of productivity in open source software development (*Vasilescu et al., 2015*). Using similar data, *Ortu et al. (2017)* demonstrate that diversity of nationality among team members is a predictor of productivity. However, they also show this demographic characteristic of a team leads to less polite and civil communication (*via* filed issues and discussion boards).

*Nafus (2012)*'s early qualitative study of gender in open source, through interview and discourse analysis of patch notes, describes sexist behavior that is linked to low participation and tenure for women in distributed software projects.

The impact of codes of conduct (CoC) - which provide formal expectations and rules for the behavior of participants in a software project - have been studied in a variety of settings. In open-source software projects codes of conduct have been shown to be widely reused (*e.g.*, Ubuntu, Contributor Covenant, Django, Python, Citizen, Open Code of Conduct, and Geek Feminism have been reused more than 500 times by projects on GitHub) (*Tourani, Adams & Serebrenik, 2017*).

There are few studies of the role and use of codes of conduct in research software development. *Joppa et al. (2013)* point to the need for developing rules which govern multiple aspects of scientific software development, but specific research that addresses the prevalence, impact, and use of a code of conduct in research software development have not been previously reported.

The 2018 survey of the research software engineer community across seven countries (*Philippe et al., 2019*) showed the percentage of respondents who identified as male as between 73% (US) and 91% (New Zealand). Other diversity measures are country-specific and were only collected in the UK and US, but in both, the dominant group is overrepresented compared with its share of the national population.

Therefore, we pose the following research question to guide our specific survey questions related to diversity –*RQ7: How do current Research Software Projects document diversity statements or codes of conduct, and what support is needed to further diversity initiatives?*

# METHODS

To understand sustainability issues related to the development and use of research software, we developed a Qualtrics (http://www.qualtrics.com) survey focused on the seven research questions defined in the Background section. This section describes the design of the survey, the solicited participants, and the qualitative analysis process we followed.

## Survey design

We designed the survey to capture information about how individuals develop, use, and sustain research software. The survey first requested demographic information to help us characterize the set of respondents. Then, we enumerated 38 survey questions (35 multiple choice and 3 free response). We divided these questions among the seven research questions defined in the Background Section. This first set of 38 questions went to all survey participants, who were free to skip any questions.

Then, to gather more detailed information, we gave each respondent the option to answer follow-up questions on one or more of the seven topic areas related to the research questions. For example, if a respondent was particularly interested in *Development Practices* she or he could indicate their interest in answering more questions about that topic. Across all seven topics, there were 28 additional questions (25 multiple choice and 3 free response). Because the follow-up questions for a particular topic were only presented to respondents who expressed interest in that topic, the number of respondent to these questions is

significantly lower than the number of respondents to first set of 38 questions. This discrepancy in the number of respondents is reflected in the data presented below.

In writing the questions, where possible, we replicated the wording of questions from the previous surveys about research software (described in the Background Section). In addition, because we assumed that respondents would be familiar with the terms used in the survey and to simplify the text, we did not provide definitions of terms in the survey itself.

## Survey participants

We distributed the survey to potential respondents through two primary venues:

1. **Email Lists**: To gather a broad range of perspectives, we distributed the survey to 33,293 United States NSF and 39,917 United States NIH PIs whose projects were funded for more than $200K in the five years prior to the survey distribution and involve research software and to mailing lists of research software developers and research software projects.

2. **Snowball Sampling**: We also used snowballing by asking people on the email lists to forward the survey to others who might be interested. We also advertised the survey *via* Twitter.

The approach we used to recruit participants makes it impossible to calculate a response rate. We do not know how many times people forwarded the survey invitation or the number of potential participants reached by the survey.

## Research ethics

We received approval for the survey instruments and protocols used in this study from the University of Notre Dame Committee Institutional Review Board for Social and Behavioral Responsible Conduct of Research (protocol ID 18-08-4802). Prior to taking the survey, respondents had to read and consent to participate. If a potential respondent did not consent, the survey terminated. To support open science, we provide the following information: (1) the full text of the survey and (2) a sanitized version of the data (*Carver et al., 2021*). We also provide a link to the scripts used to generate the figures that follow (*Carver et al., 2022*). Qualtrics collected the IP address and geo location from survey respondents. We removed these columns from the published dataset. However, we did not remove all comments that might lead people to make educated guesses about the respondents.

## ANALYSIS

After providing an overview of the participant demographics, we describe the survey results relative to each of the research questions defined in the Background section. Because many of the survey questions were optional and because the follow-up questions only went to a subset of respondents, we report the number of respondents for each question along with the results below. To clarify which respondents received each question, we provide some text around each result. In addition, when reporting results from a follow-up question, the text specifically indicates that it is a follow-up question and the number of respondents will be much smaller.

### Participant demographics

We use each of the key demographics gathered on the survey to characterize the respondents (*e.g.*, the demographics of the sample). Note that because some questions were optional, the number of respondents differs across the demographics.

#### Respondent type

We asked each respondent to characterize their relationship with research software as one of the following:

- *Researcher* - someone who only uses software
- *Developer* - someone who only develops software
- *Combination* - both of the above roles

The respondents were fairly evenly split between *Researchers* at 43% (473/1109) and *Combination* at 49% (544/1109), with the remaining 8% (92/1109) falling into the *Developer* category. Note that depending on how the respondent answered this question, they received different survey questions. If a respondent indicated they were a *Researcher*, they did not receive the more development-oriented questions. For the remainder of this analysis, we use these subsets to analyze the data. If the result does not indicate that it is describing results from a subset of the data, then it should be interpreted as being a result from everyone who answered the question.

#### Organization type

Next, respondents indicated the type of organization for which they worked. The vast majority 86% (898/1048) worked for *Educational Institutions*. That percentage increased to 93% (417/447) for Researcher type respondents

#### Geographic location

Because the focus of the URSSI project is the United States, we targeted our survey to US-based lists. As a result, the vast majority of responses (990/1038) came from the United States. We received responses from 49 states (missing only Alaska), plus Washington, DC, and Puerto Rico.

#### Job title

People involved in developing and using research software have various job titles. For our respondents, *Faculty* was the most common, given by 63% (668/1046) of the respondents and 79% (354/447) of the Researcher type respondents. No other title was given by more then 6% of the respondents.

#### Respondent age

Overall, 77% (801/1035) of the respondents are between 35 and 64 years of age. The percentage is slightly higher for Researcher type respondents (370/441–84%) and slightly lower for Combination type respondents (378/514–74%).

#### Respondent experience

The respondent pool is highly experienced overall, with 77% (797/1040) working in research for more than 10 years and 39% (409/1040) for more than 20 years. For the

**Table 2** Disciplines of respondents.

| Discipline | Total | Researchers | Developers | Combination |
|---|---|---|---|---|
| Biological sciences | 325 | 149 | 20 | 156 |
| Mathematical and physical sciences | 480 | 174 | 36 | 269 |
| Engineering | 207 | 74 | 18 | 115 |
| Computer & information science | 268 | 61 | 33 | 174 |
| Medicine | 105 | 49 | 5 | 51 |
| Dentistry and health | 17 | 8 | 1 | 8 |
| Social sciences | 98 | 55 | 2 | 41 |
| Humanities and language based studies | 24 | 4 | 1 | 18 |
| Administrative & business studies | 12 | 10 | 0 | 2 |
| Agricultural | 23 | 10 | 0 | 13 |
| Forestry and veterinary science | 11 | 5 | 1 | 5 |
| Education | 61 | 29 | 2 | 30 |
| Architecture and planning | 7 | 1 | 1 | 5 |
| Design | 12 | 4 | 1 | 7 |
| Creative & performing arts | 7 | 0 | 0 | 7 |

Researcher type respondents, those numbers increase to 84% (373/444) with more than 10 years and 44% (197/444) with more than 20 years.

### Gender

In terms of self-reported gender, 70% (732/1039) were Male, 26% (268/1039) were Female, with the remainder reporting Other or Prefer not to say. For Researcher respondents, the percentage of Females is higher (151/443–34%).

### Discipline

The survey provided a set of choices for the respondents to choose their discipline(s). Respondents could choose more than one discipline. Table 2 shows the distribution of respondents by discipline. Though the respondents represent a number of research disciplines, our use of NSF and NIH mailing lists likely skewed the results towards participants from science and engineering fields.

## Software engineering practices

This section focuses on answering *RQ1: What activities do research software developers spend their time on, and how does this impact the perceived quality and long-term accessibility of research software?*

### Where respondents spend software time

We asked respondents what percentage of their time they currently spend on a number of software activities and what percentage of time they would ideally like to spend on those activities. The box and whisker plots in Fig. 1 shows there is a mismatch between these two distributions, *SpentTime* and *IdealSpentTime*, respectively. Overall, respondents would like to spend more time in *design* and *coding* and less time in *testing* and *debugging*. However, the differences are relatively small in most cases.

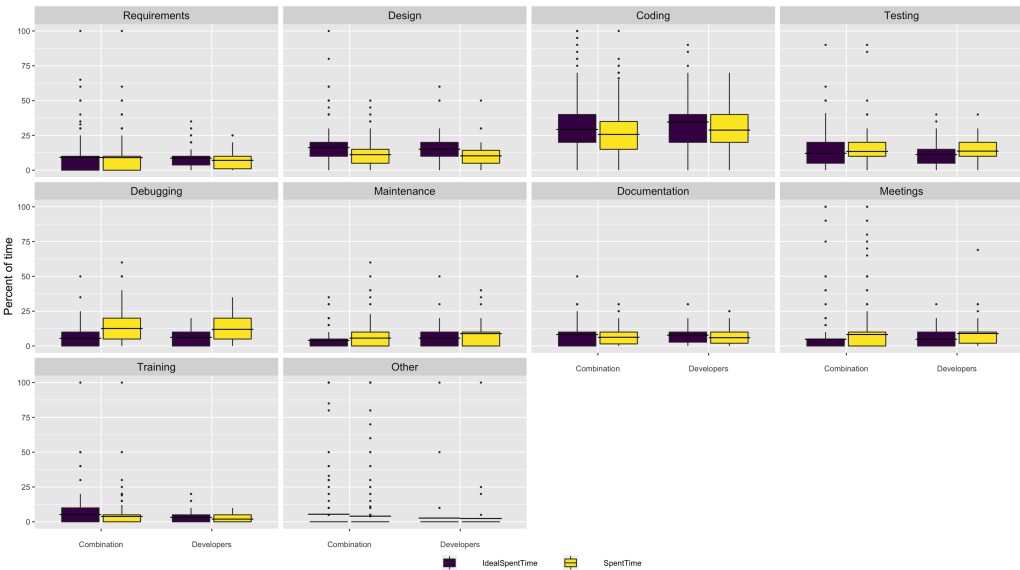

**Figure 1  Where respondents spend software time.** The dots represent outliers.

### Which aspects of the software development process are more difficult than they should be?

Figure 2 illustrates the results for this follow-up question. This question was multi-select where respondents could choose as many answers as were appropriate. Interestingly, the aspects most commonly reported are those that are more related to people issues rather than to technical issues (*e.g.*, *finding personnel/turnover*, *communication*, *use of best practices*, *project management*, and *keeping up with modern tools*). The only ones that were technical were *testing* and *porting*.

### Use of testing

Focusing on one of the technical aspect that respondents perceived to be more difficult than it should be, we asked the respondents how frequently they employ various types of testing, including: Unit, Integration, System, User, and Regression. The respondents could choose from *frequently*, *somewhat*, *rarely*, and *never*. Figure 3 shows the results from this question. The only type of testing more than 50% of respondents used frequently was **Unit testing** (231/453–51%). On the other extreme only about 25% reported using **System** (118/441–27%) or **Regression** testing (106/440–24%) frequently.

### Use of open-source licensing

Overall 74% (349/470) of the respondents indicated they used an open-source license. This percentage was consistent across both combination and developer respondents. However, this result still leaves 26% of respondents who do not release their code under an open-source license.

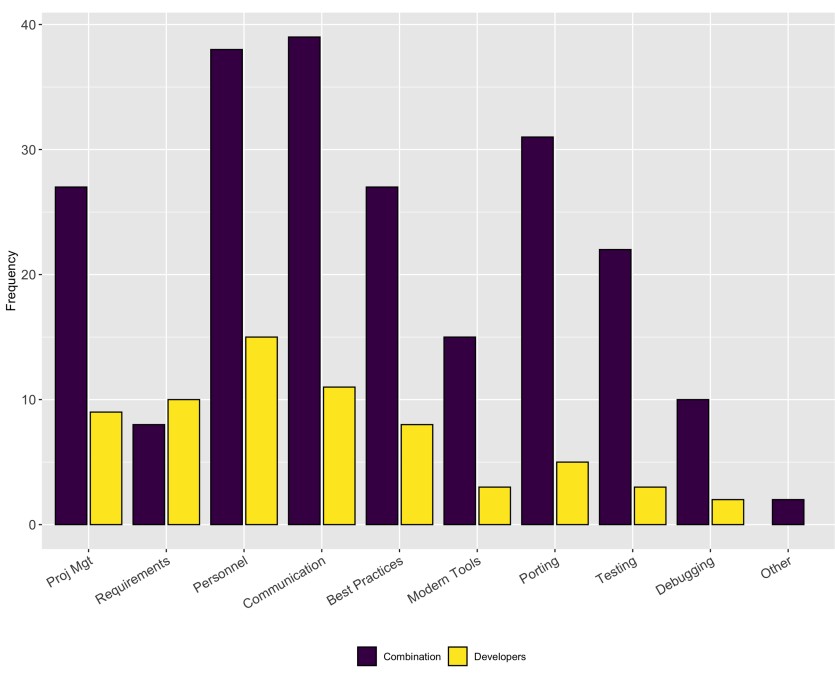

**Figure 2  Aspects of software development that are more difficult than they should be.**

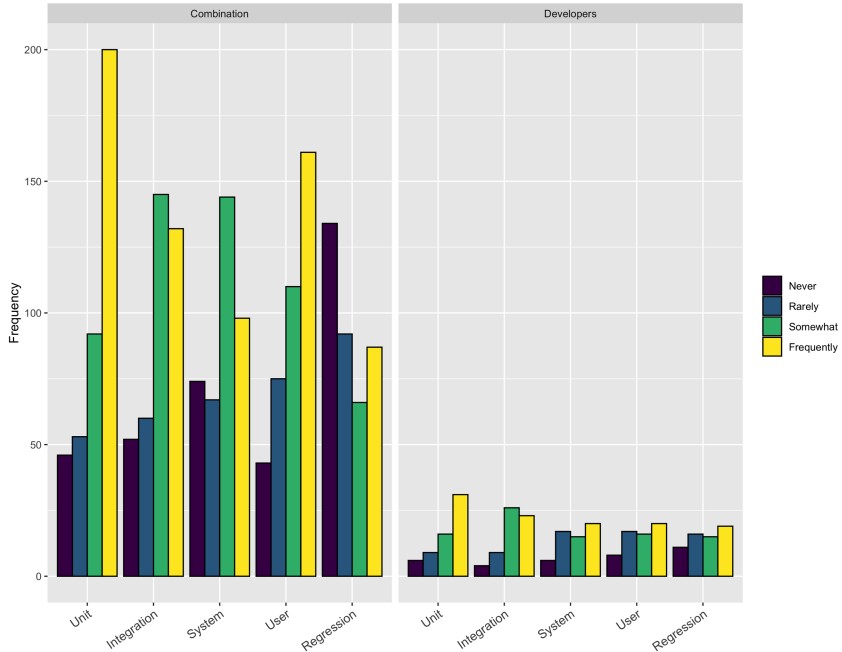

**Figure 3  Use of testing.**

### Frequency of using *best* practices

As a follow-up question, we asked the respondents how frequently they used a number of standard software engineering practices. The response options were *Never*, *Sometimes*, *Half of the time*, *Most of the time*, *Always*. The following list reports those who responded *Most of the time* or *Always* for the most commonly used practices (in decreasing order):

- Continuous Integration – 54% (54/100)
- Use of coding standards – 54% (54/100)
- Architecture or Design – 51% (52/101)
- Requirements – 43% (43/101)
- Peer code review – 34% (34/99)

### Documentation

As a follow-up question, in terms of what information respondents document, only 55% (56/101) develop **User manuals or online help** either *Most of the time* or *Always*. However, 86% (87/101) **Comment code** and 95% (96/101) **Use descriptive variable/method names** either *Most of the time* or *Always*. Interestingly, even though a very large percentage of respondents indicated that they comment their code, when we look in more detail at the other types of information documented, we see a different story. The following list reports those who responded *Most of the time* or *Always* for each type of documentation:

- Requirements – 49% (49/100)
- Software architecture or design – 34% (34/100)
- Test plans or goals – 25% (25/100)
- User stories/use cases – 24% (24/100)

## Tools

This section focuses on answering *RQ2: What tools do research software developers use and what additional tools are needed to support sustainable development practices?*

### Tools support for development activities

The results in Fig. 4 indicate that a large majority of respondents (340/441–77%) believe **Coding** is *Extremely supported* or *Very supported* by existing tools. Slightly less than half of the respondents find **Testing** (196/441–44%) and **Debugging** (188/441–43%) to be *Extremely supported* or *Well supported*. Less than 30% of the respondents reported **Requirements**, **Architecture/design**, **Maintenance**, and **Documentation** as being well-supported. Because coding is the only practice where more than half of the respondents indicate *Extremely supported* or *Very supported*, these responses indicate a clear opportunity for additional (or better) tool support in a number of areas.

### Version control and continuous integration

In a follow-up question, almost all of those who responded (83/87) indicated they do use version control. In addition, a slightly lower but still very large percentage of respondents (74/83) indicated they used Git either *Always* or *Most of the time*. Git was by far the most commonly used version control system. Finally, almost all respondents (76/83) check

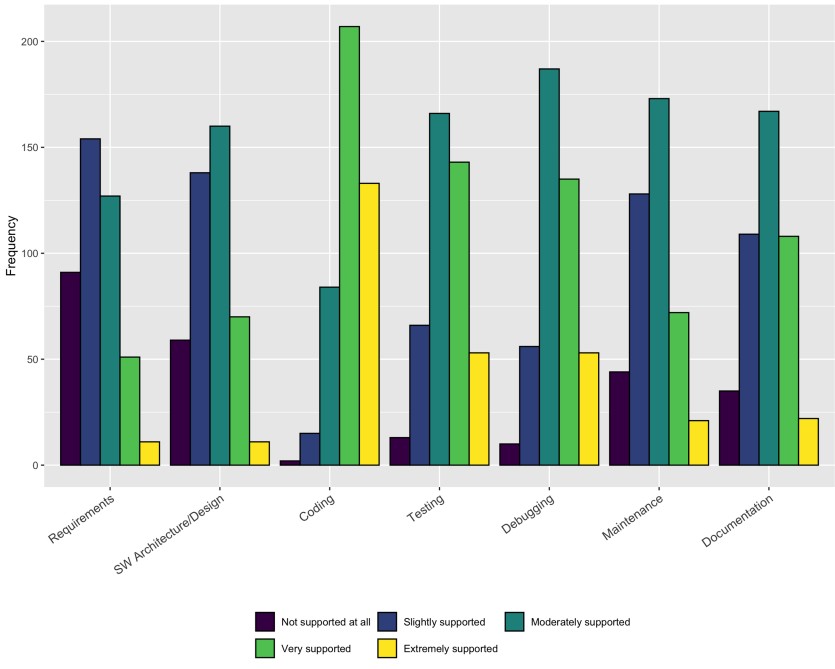

**Figure 4  Availability of tool support.**

their code into the version control system either *after every change* or *after a small group of changes*. However, when we investigate further about the version control practices, we find 29% of the Combination type respondents (26/56) indicated they use *copying files to another location* and 10% (6/56) used *zip file backups* as their method of version control either *Always* or *Most of the time*. While many respondents do use a standard version control system, the large number of Combination respondents who rely on *zip file backups* suggests the use of standard version control methods is an area where additional training could help. Regarding continuous integration, less than half of the respondents (39/84) indicated they used it either *Always* or *Most of the time*. This result suggests another area where additional training could help.

## Training

This section focuses on answering RQ3: *What training is available to research software developers and does this training meet their needs?*

### *Have you received training?*

The percentage of respondents answering *yes* depends upon the type of respondent: Developers - 64% (39/61), Combination - 44% (170/384), and Researchers - 22% (93/421). When we examine the responses to this question by gender we also see a difference: 30% (69/229) of the Female respondents received training compared with 37% (220/601) of the Male respondents.

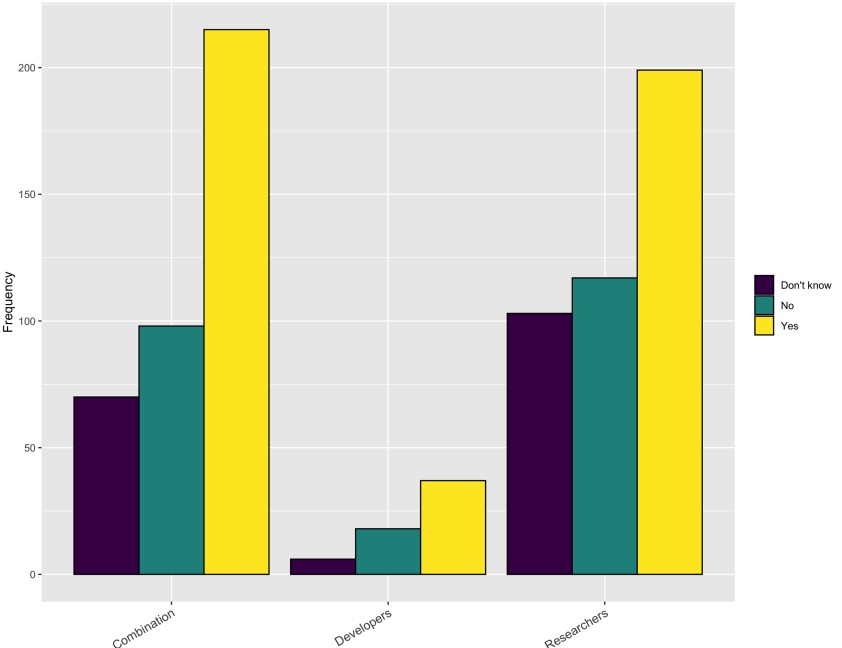

**Figure 5  Sufficient opportunities for training.**

### Where have you received training?

Of those who reported receiving training, 82% had received this training from a *Class/School* and 48% received it *Online/Self-directed*. (Note that respondents could report more than one source of training.) Interestingly, only 10% of the respondents had *Software Carpentry* training. When we examine these responses by gender, we find approximately the same percentage of Male and Female responds received training in a *Class/School* or *Online/Self-directed*. However, the percentage of Female respondents who reported *Software Carpentry* or *Other* training was lower than for Male respondents.

### Are there sufficient opportunities for training?

When we turn to the availability of relevant training opportunities, an interesting picture emerges. As Fig. 5 shows, slightly more than half of the respondents indicate there is sufficient training available for obtaining new software skills. However, when looking at the response based upon gender, there is a difference with 56% of male respondents answering positively but only 43% of the female respondents answering positively. But, as Fig. 6 indicates, approximately 75% of the respondents indicated they do not have sufficient time to take advantage of these opportunities. These results are slightly higher for female respondents (79%) compared with male respondents (73%). So, while training may be available, respondents do not have adequate time to take advantage of it.

### Preferred modes for delivery of training

The results showed that there is not a dominant approach preferred for training. Carpentries, Workshops, MOOCs, and On-site custom training all had approximately

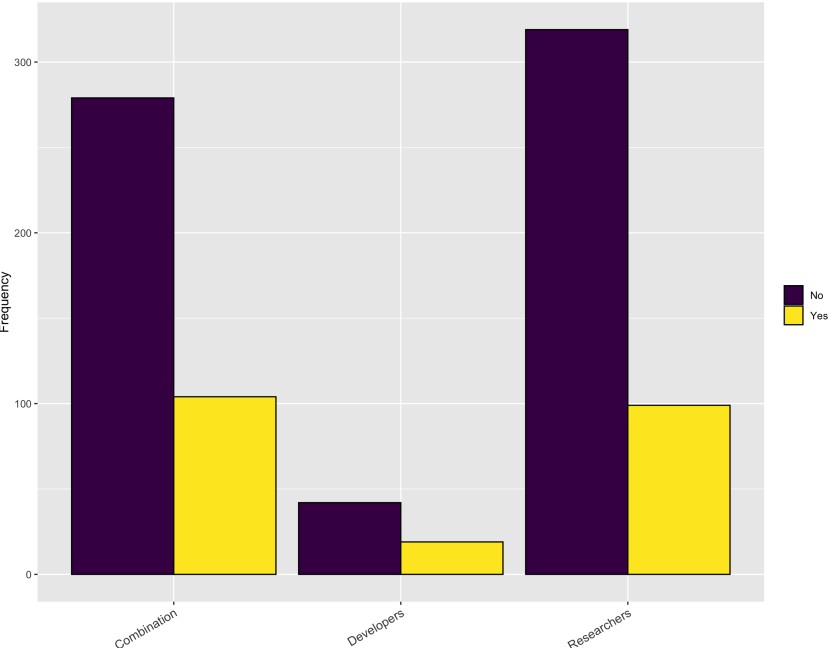

**Figure 6** Sufficient time for training.

the same preference across all three topic ares (Development Techniques, Languages, and Project Management). This result suggests that there is benefit to developing different modes of training about important topics, because different people prefer to learn in different ways.

## Funding

This section uses the relevant survey questions to answer the two research questions related to funding:

- *RQ4a: What is the available institutional support for research software development?*
- *RQ4b: What sources of institutional funding are available to research software developers?*

First, 54% (450/834) of the respondents reported they have included funding for software in their proposals. However, that percentage drops to 30% (124/408) for respondents who identify primarily as Researchers.

When looking at the specific types of costs respondents include in their proposals, 48% (342/710) include costs for *developing new software*, 22% (159/710) for *reusing existing software* and 29% (209/710) for *maintaining/sustaining software*. (Note that respondents could provide more than one response.) It is somewhat surprising to see such a large number of respondents who include funding for maintaining and sustaining software.

In examining the source of funding for the projects represented by the survey respondents, the largest funder is NSF, at 36%. But, as Fig. 7 shows, a significant portion of funding comes from the researchers' own institutions. While other funding agencies provide funding for the represented projects and may be very important for individual

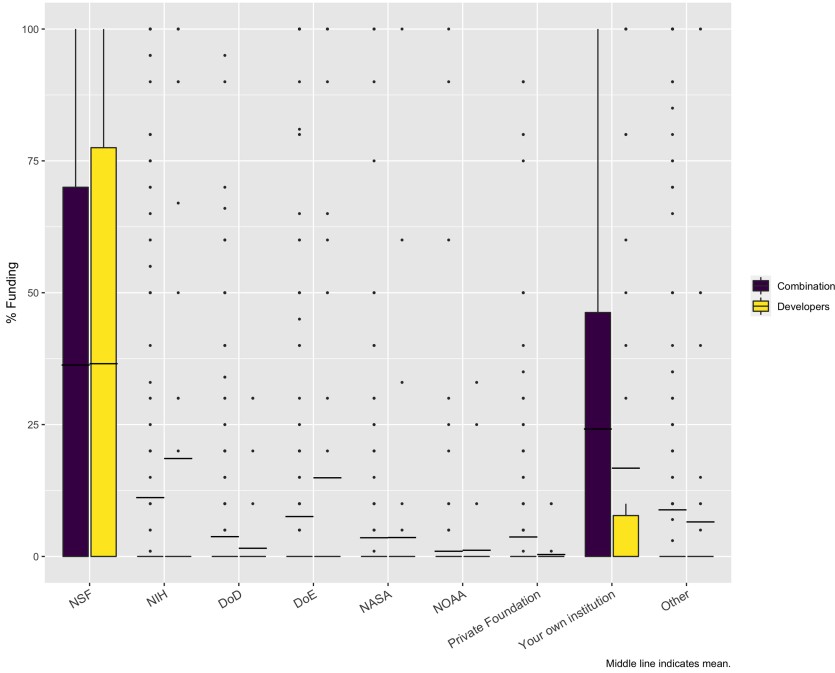

**Figure 7  Sources of funding.** Except for NSF and "Your own institution", the minimum, Q1, median, and Q3 are all zero. The dashed line is the mean for these categories. The dots are all outliers.

respondents, overall, they have little impact. This result could have been impacted by the fact that we used a mailing list of NSF projects as one means of distributing the survey. However, we also used a list of NIH PIs who led projects funded at least at $200K, so it is interesting that NSF is still the largest source.

In terms of the necessary support, Fig. 8 indicates that, while institutions do provide some RSE, financial, and infrastructure support, it is inadequate to meet the respondents' needs, overall. In addition, when asked in a follow-up question whether the respondents have sufficient funding to support software development activities for their research the overwhelming answer is no (Fig. 9).

When asked about whether current funding adequately supports some key phases of the software lifecycle, the results were mixed. Respondents answered on a scale of 1–5 from *insufficient* to *sufficient*. For *Developing new software* and *Modifying or reusing existing software* there is an relatively uniform distribution of responses across the five answer choices. However, for *Maintaining software*, the responses skew towards the *insufficient* end of the scale.

For respondents who develop new software, we asked (on a 5-point scale) whether their funding supports various important activities, including *refactoring*, *responding to bugs*, *testing*, *developing new features*, and *documentation*. In all cases less than 35% of the respondents answered 4 or 5 (*sufficient*) on the scale.

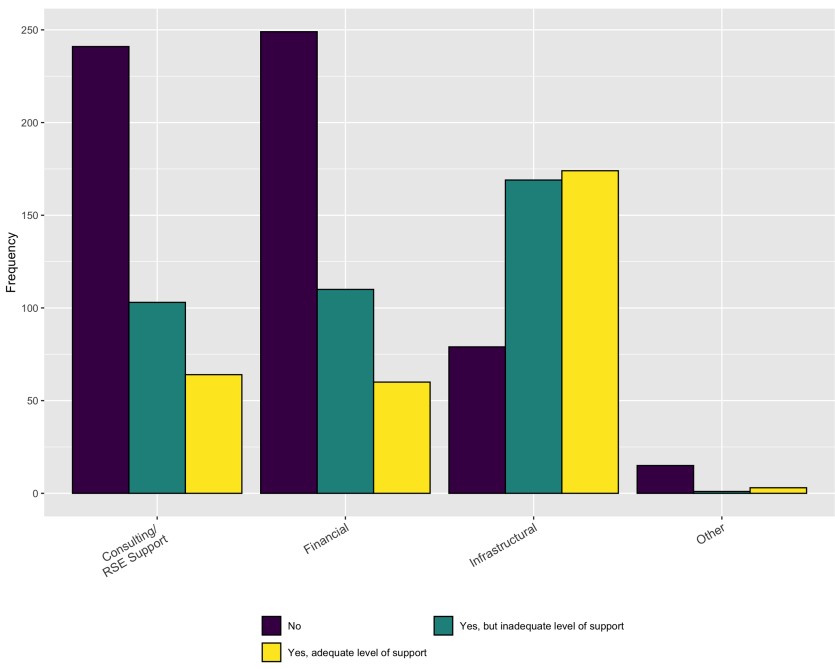

**Figure 8  Sufficiency of institutional support.**

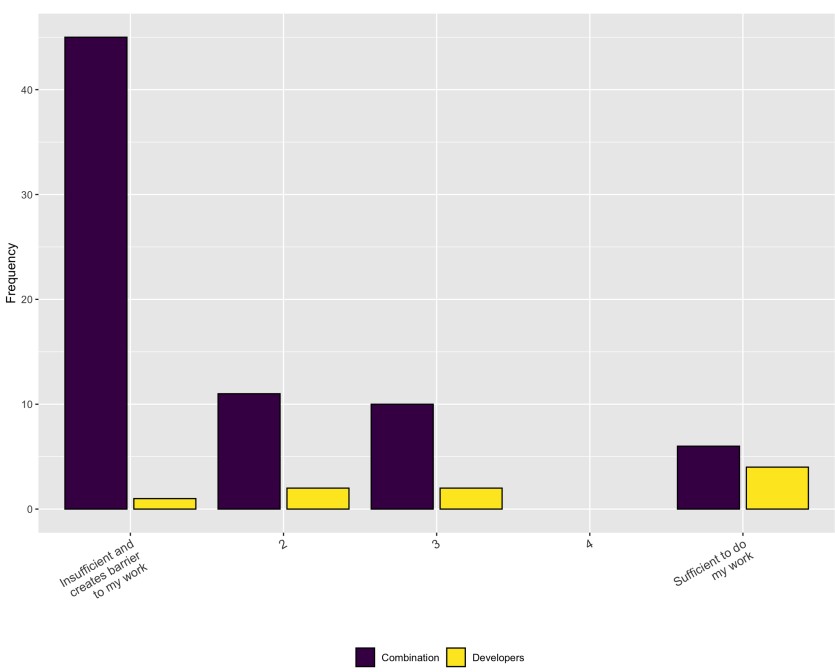

**Figure 9  Necessary funding to support software development activities.**

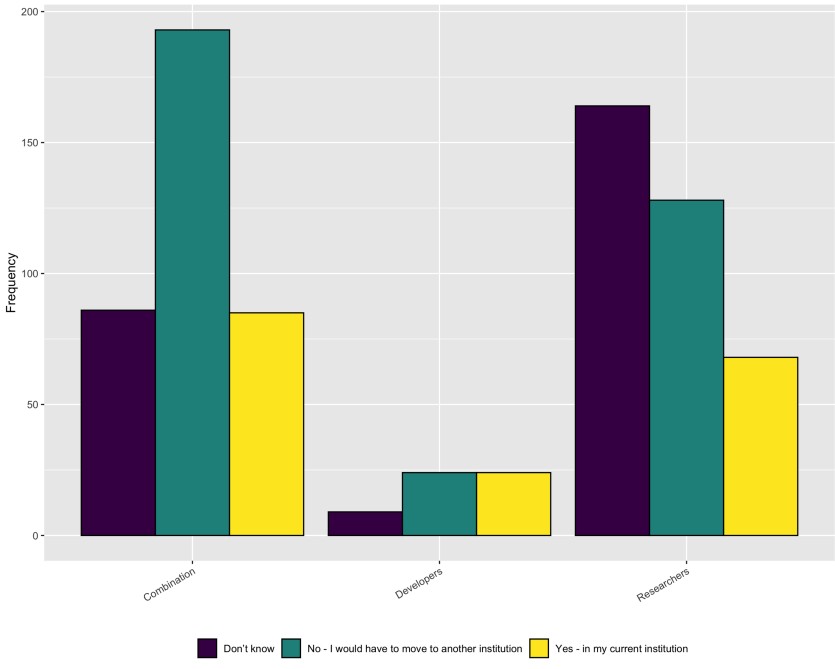

**Figure 10** **Opportunities for career advancement for software developers.**

## Career paths

This section focuses on answering *RQ5: What factors impact career advancement and hiring in research software?*

Institutions have a number of different job titles for people who develop software. The most frequently reported title is *postdoc* (411), with other titles including *Research Software Engineers* (223), *Research Programmers* (251), *Software Developer* (253), and *Programmer* (253). There were also a good number of respondents who were *Faculty* (215) or *Research Faculty* (242). Note that people could provide more than one answer, so the total exceeds the number of respondents.

While there are a number of job titles that research software developers can fill, unfortunately, as Fig. 10 shows, the respondents saw little chance for career advancement for those whose primary role is software development. Only 21% (153/724) of the *Combination* and *Researcher* respondents saw opportunity for advancement. The numbers were slightly better at 42% (24/57) for those who viewed themselves as *Developers*. When we look at the result by gender, only 16% (32/202) of the female respondents see an opportunity for advancement compared with 24% (139/548) for the male respondents.

We asked respondents who have the responsibility for hiring people into software development roles to indicate the importance of the following concerns:

- Identifying a pipeline for future staff
- Attracting staff from underrepresented groups
- Ability of staff to work across disciplines
- Competing with industry for high performers

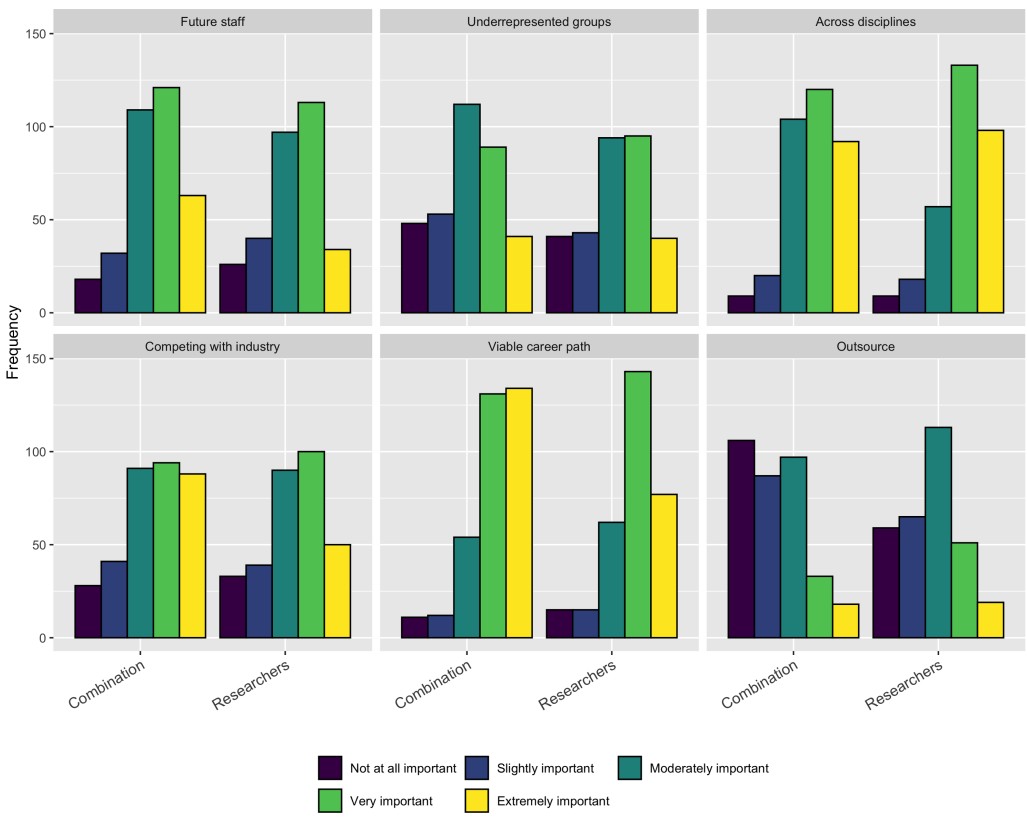

**Figure 11  Concerns when trying to hire software development staff.**

- Offering a viable career path
- Opportunities to outsource skilled work

As Fig. 11 shows, most factors were at least moderately important, with the exception of opportunities to outsource work.

When examining the perceptions of those that have been hired into a software development role, we asked a similar question. We asked respondents the importance of the following concerns when they were hired into their current role:

- Diversity in the organization
- Your experience as a programmer or software engineer
- Your background in science
- Your knowledge of programming languages
- Your knowledge and capabilities across disciplines
- Your potential for growth

As Fig. 12 shows, for job seekers, their background in science, ability to work across disciplines, and opportunities for growth were the most important factors. Interestingly,

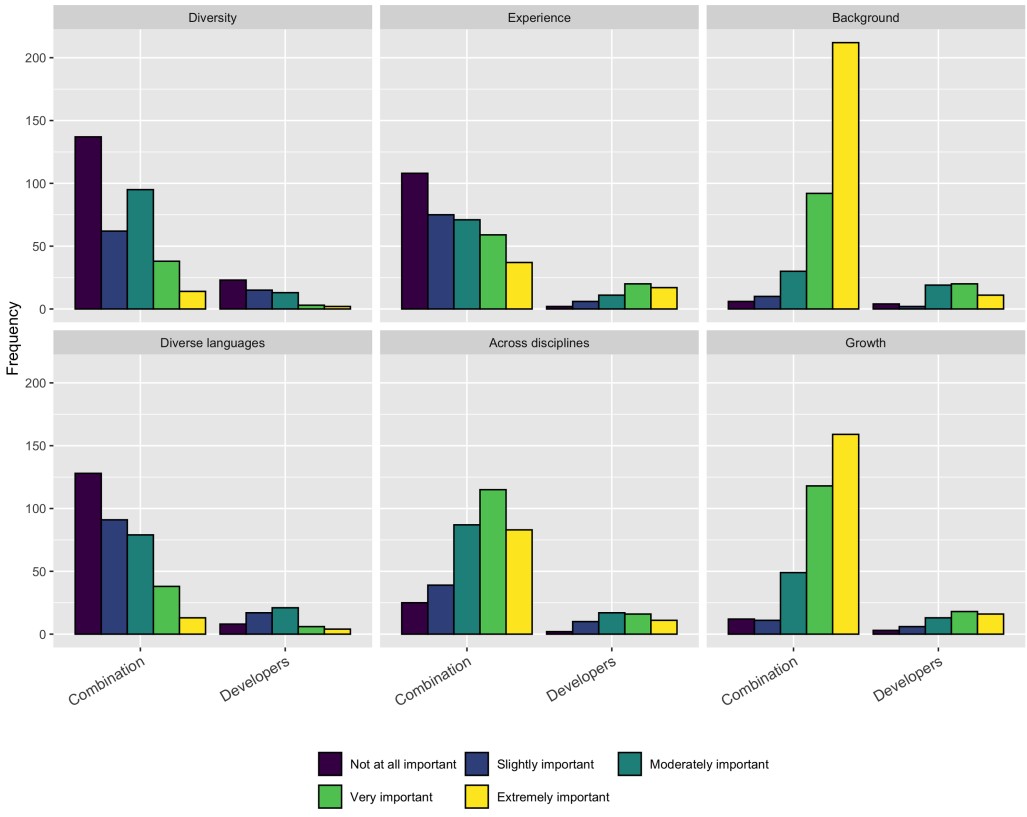

**Figure 12** **Concerns when being hired as software development staff.**

the respondents saw less importance in their experience as a programmer or software engineering or their knowledge of programming languages.

Besides these factors, we asked a follow-up question about the importance of the following factors when deciding on whether to accept a new job or a promotion:

- Title of the position
- Salary raise
- Responsibilities for a project or part of a project
- Leading a team
- Available resources such as travel money

As Fig. 13 shows *Salary*, *Responsibility*, *Leadership*, and *Resources* are the most important factors respondents consider when taking a job or a promotion.

Lastly, in terms of recognition within their organization, in a follow-up question, we asked respondents to indicate whether other people in their organization use their software and whether other people in their organization have contacted them about developing software. Almost everyone (54/61) that responded to these follow-up questions indicated that other people in the organization use their software. In addition just over half of the

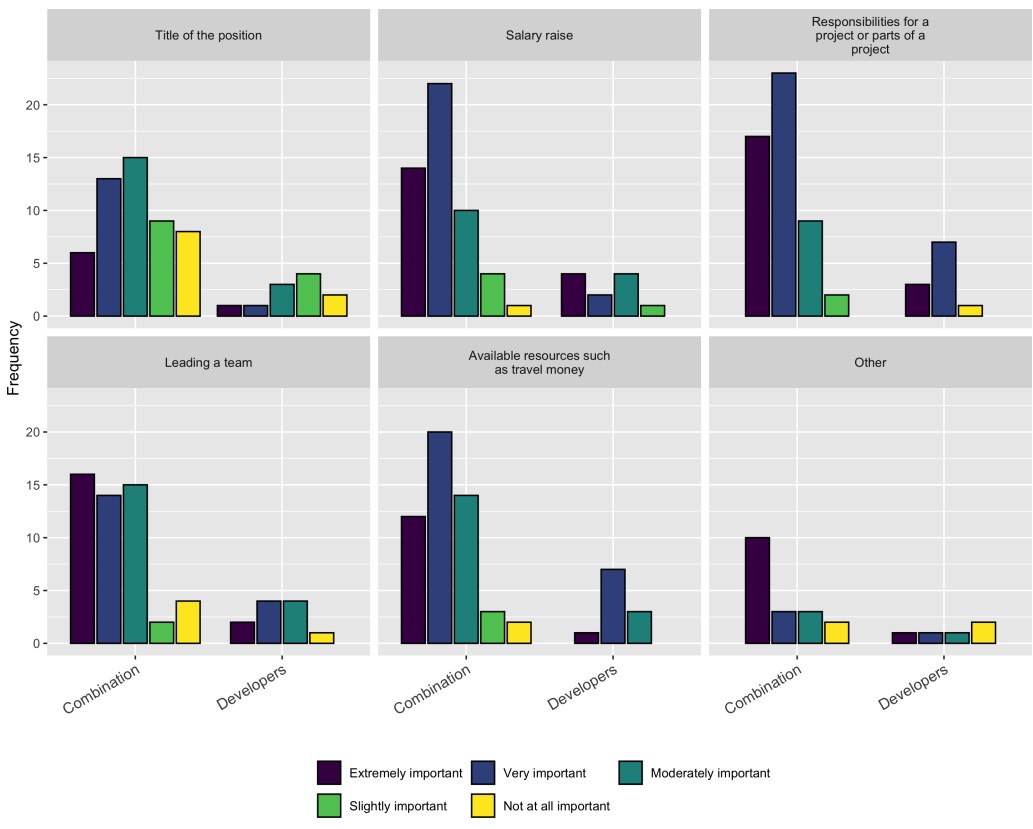

**Figure 13** **Importance of factors when taking a job or promotion.**

Combination respondents (27/51) and 72% (8/11) of the Developer respondents indicated that people in their organization had contacted them about writing software for them.

## Credit

This section uses the relevant survey questions to answer the two research questions related to credit:

- *RQ6a: What do research software projects require for crediting or attributing software use?*
- *RQ6b: How are individuals and groups given institutional credit for developing research software?*

When asked how respondents credit software they use in their research, as Fig. 14 shows, the most common approaches are either to *cite a paper about the software* or to *mention the software by name*. Interestingly, authors tended to *cite the software archive itself*, *mention the software URL*, or *cite the software URL* much less frequently. Unfortunately, this practice leads to fewer trackable citations of the software, making it more difficult to judge its impact.

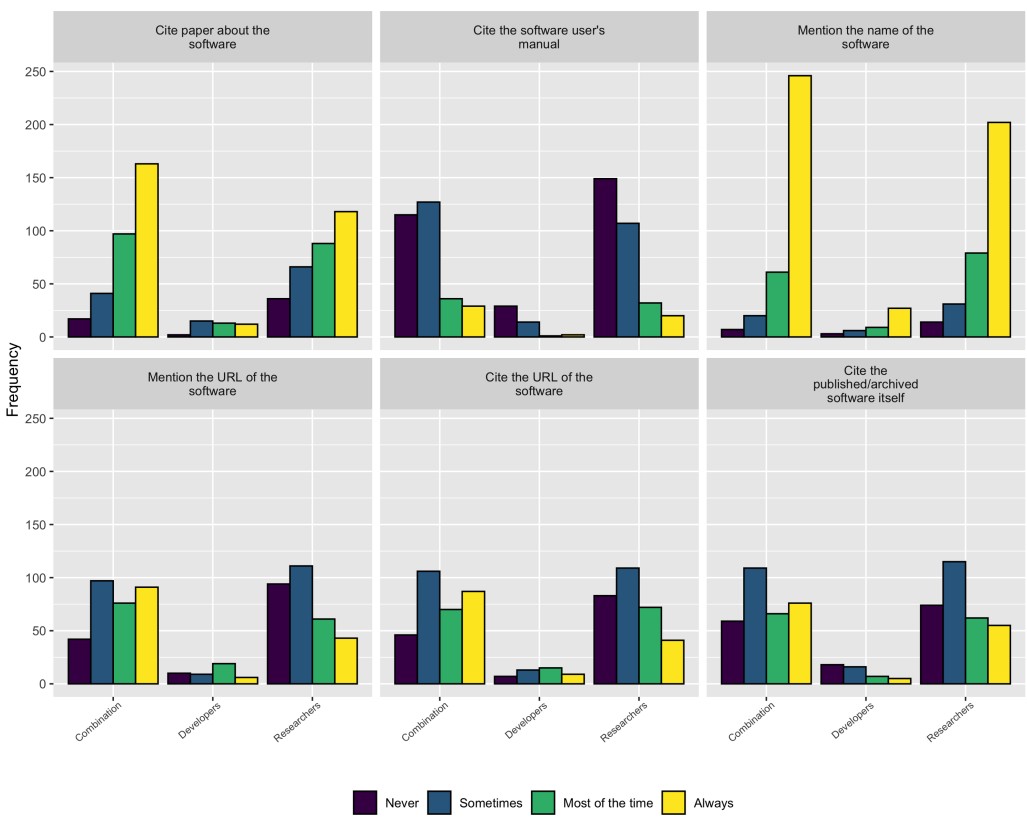

**Figure 14** **How authors credit software used in their research.**

Following on this trend of software work not being properly credited, when asked how they currently receive credit for their own software contributions, as Fig. 15 shows, none of the standard practices appear to be used very often.

An additional topic related to credit is whether respondent's contributions are valued for performance reviews or promotion within their organization. As Fig. 16 shows, approximately half of the respondents indicate that software contributions are considered. Another large percentage say that it *depends*.

While it is encouraging that a relatively large percentage of respondents' institutions consider software during performance reviews and promotion some or all of the time, the importance of those contributions is still rather low, especially for respondents who identify as *Researchers*, as shown in Fig. 17.

## Diversity

This section focuses on answering *RQ7: How do current Research Software Projects document diversity statements and what support is needed to further diversity initiatives?*

When asked how well their projects recruit, retain, and include in governance participants from underrepresented groups, only about 1/3 of the respondents thought they did an "Excellent" or "Good" job. Interestingly, when asked how well they promote a culture of inclusion, 68% of the respondents (390/572) indicated they did an "Excellent"

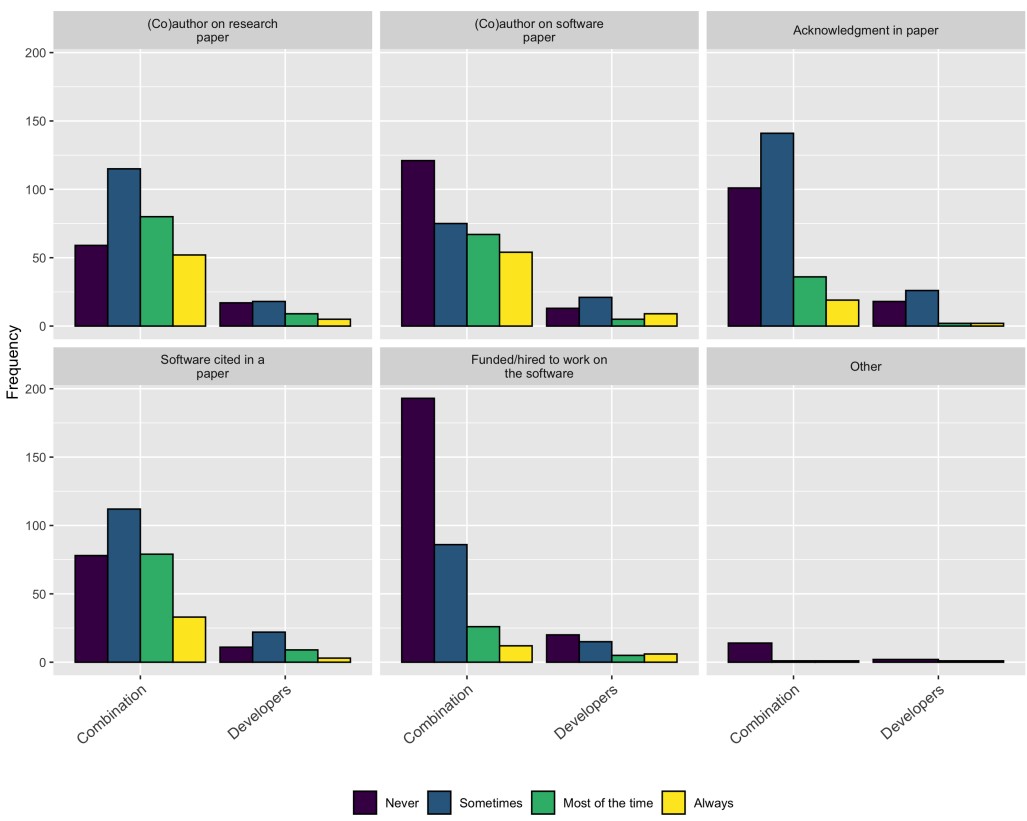

**Figure 15** **How respondents currently receive credit for their software contributions.**

or "Good" job. These two responses seem to be at odds with each other, suggesting that perhaps projects are not doing as well as they think they are. Conversely, it could be that projects do not do a good job of recruiting diverse participants, but do a good job of supporting the ones they do recruit. Figure 18 shows the details of these responses.

We asked follow-up questions about whether the respondents' projects have a diversity/inclusion statement or a code of conduct. As Figs. 19 and 20 show, most projects do not have either of these, nor do they plan to develop one. This answer again seems at odds with the previous answer that most people thought their projects fostered a culture of inclusions. However it is possible that projects fall under institutional codes of conduct or have simply decided that a code of conduct is not the best way to encourage inclusion.

Lastly, in a follow-up question we asked respondents to indicate the aspects of diversity or inclusion for which they could use help. As Fig. 21 shows, respondents indicated they needed the most help with recruiting, retaining, and promoting diverse participants. They also need help with developing diversity/inclusion statements and codes of conduct.
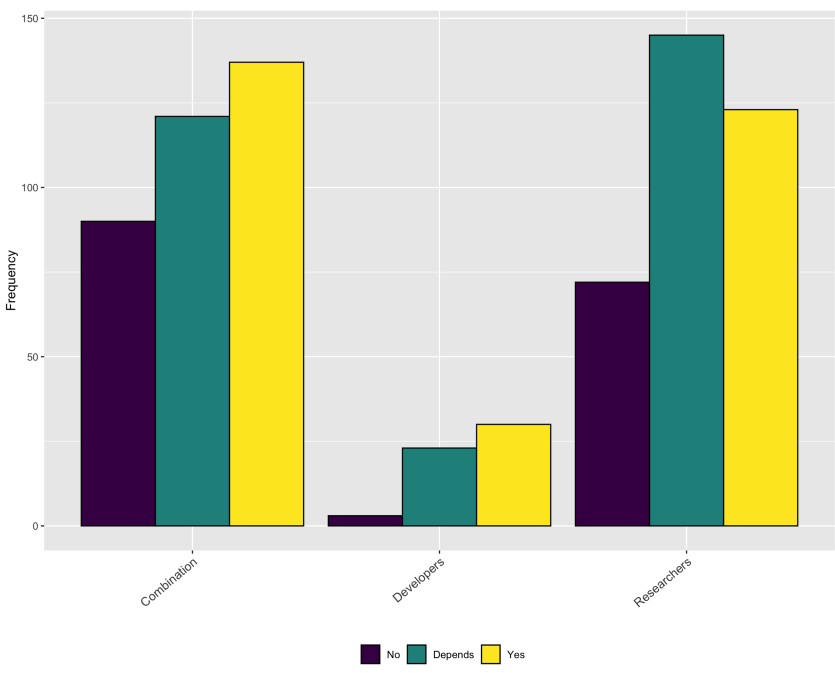

**Figure 16** Does institution consider software contributions in performance reviews or promotion cases?

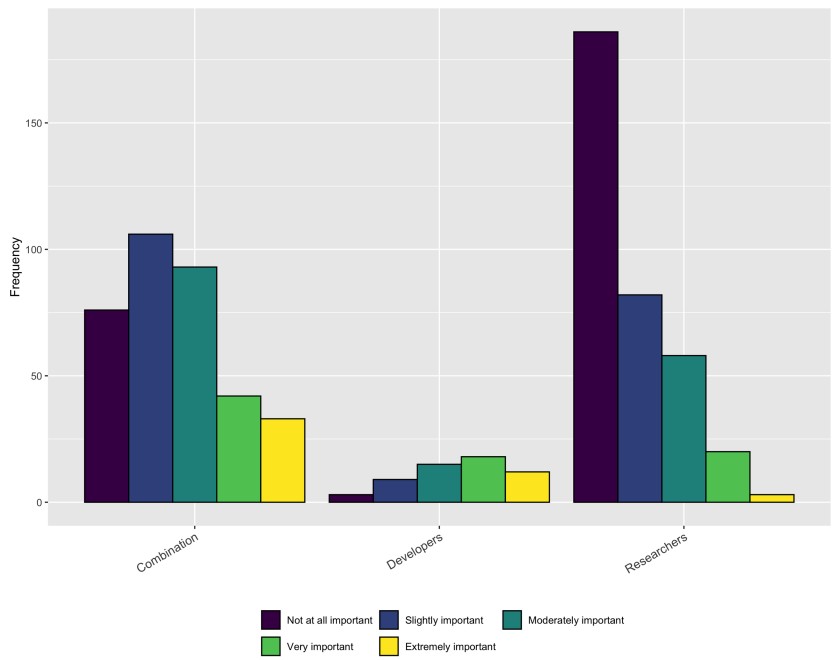

**Figure 17** Importance of software contributions during performance review or promotion.

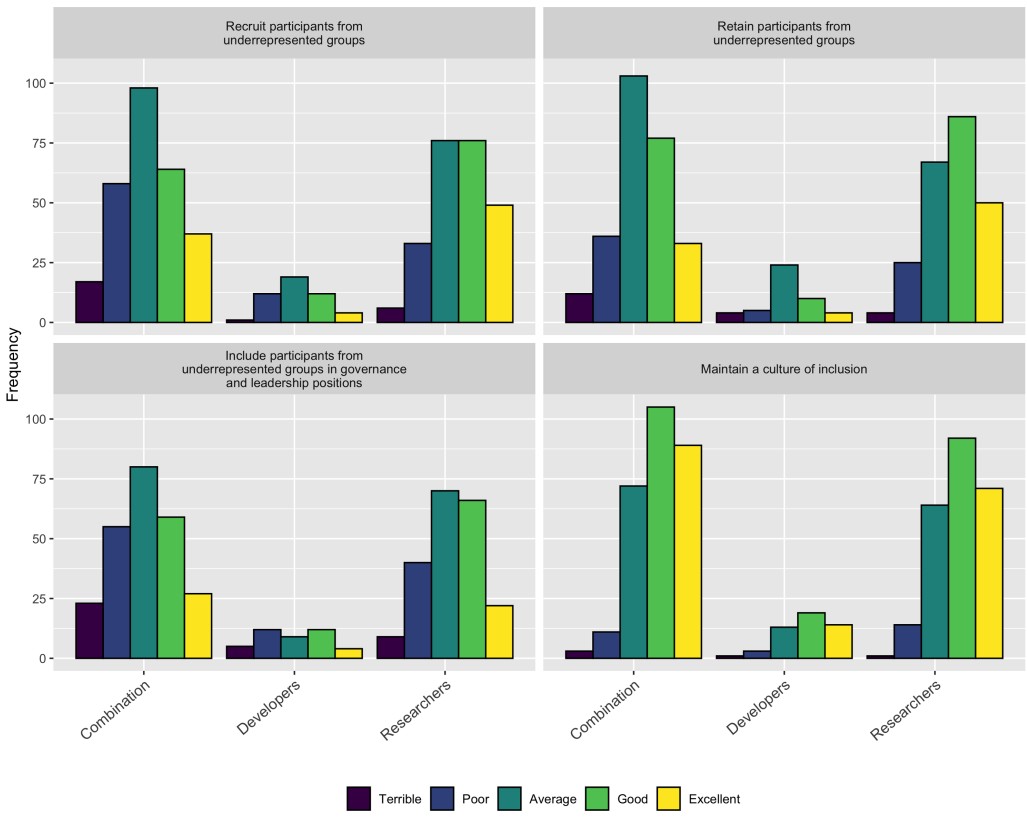

**Figure 18  How respondents think their project does with inclusion.**

# THREATS TO VALIDITY

To provide some context for these results and help readers properly interpret them, this section describes the threats to validity and limits of the study. While there are multiple ways to organize validity threats, we organize ours in the following three groups.

## Internal validity threats

Internal validity threats are those conditions that reduce the confidence in the results that researchers can draw from the analysis of the included data.

A common internal validity threat for surveys is that the data is self-reported. Many of the questions in our survey rely upon the respondent accurately reporting their perception of reality. While we have no information that suggests respondents were intentionally deceptive, it is possible that their perception about some questions was not consistent with their reality.

A second internal validity threat relates to how we structured the questions and which questions each respondent saw. Due to the length of the survey and the potential that some questions may not be relevant for all respondents, we did not require a response to all questions. In addition, we filtered out questions based on the respondent type (Researcher, Combination, or Developer) when those questions were not relevant. Last,

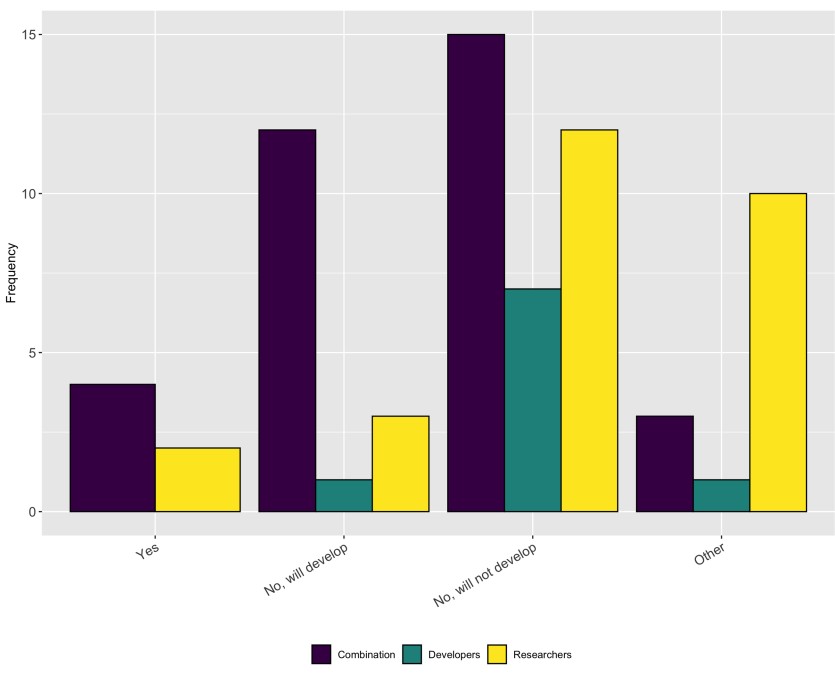

**Figure 19** **Whether the respondents' projects have a diversity/inclusion statement.**

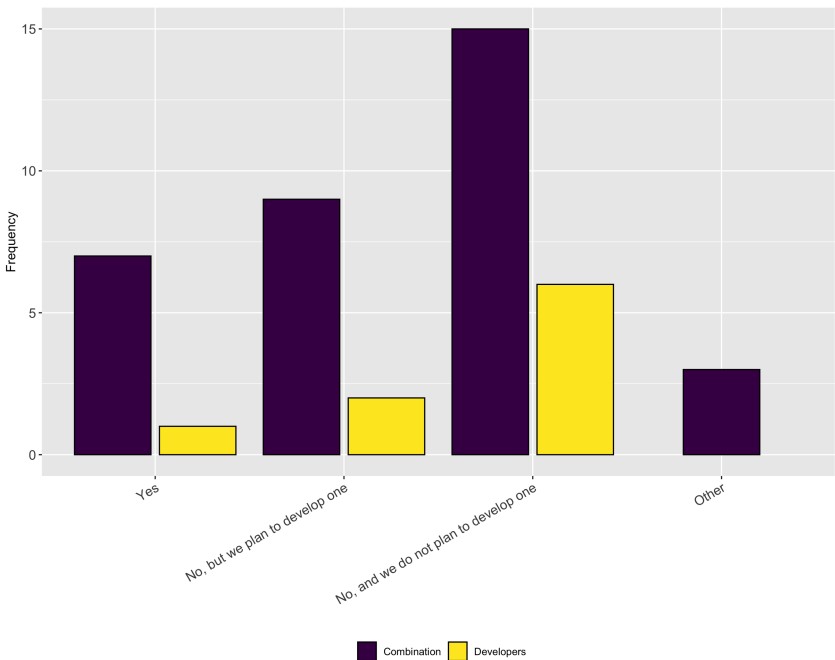

**Figure 20** **Whether the respondents' projects have a code of conduct.**

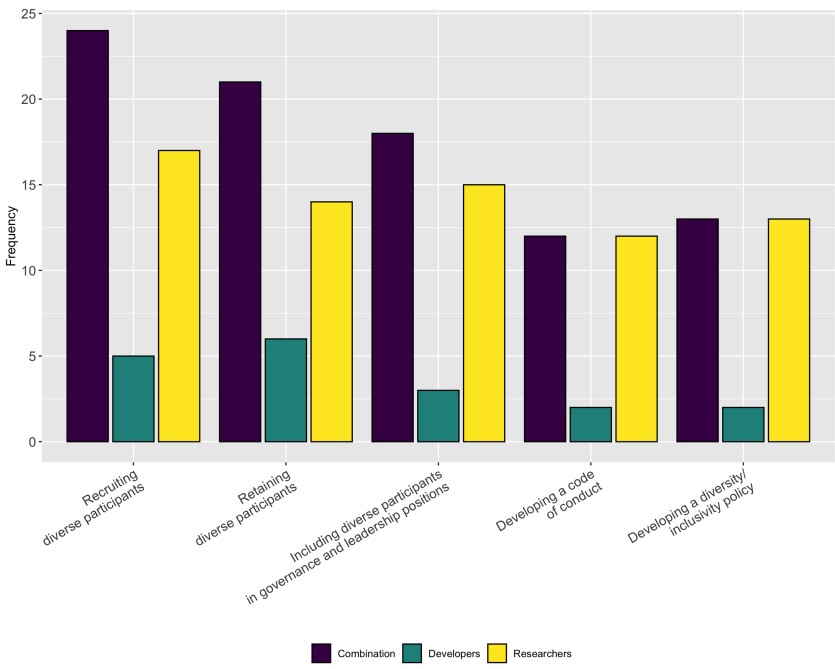

**Figure 21    Type of support needed for inclusion and diversity.**

for each topic area, we included a set of optional questions for those who wanted to provide more information. These optional questions were answered by a much smaller set of respondents. Taken together, these choices mean that the number of respondents to each question varies. In the results reported above, we included the number of people who answered each question.

## Construct validity threats

Construct validity threats describe situations where there is doubt in the accuracy of the measurements in the study. In these cases, the researchers may not be fully confident that the data collected truly measures the construct of interest.

In our study, the primary threat to construct validity relates to the respondents' understanding of key terminology. The survey used a number of terms related to the research software process. It is possible that some of those terms may have been unfamiliar to the respondents. Because of the length of the survey and the large number of terms on the survey, we chose not to provide explicit definitions for each concepts. While we have no evidence that raises concerns about this issue, it is possible that some respondents interpreted questions in ways other than how we originally intended.

## External validity threats

External validity threats are those conditions that decrease the generalizablity of the results beyond the specific sample included in the study. We have identified two key external validity threats.

The first is general sampling bias. We used a convenience sample for recruiting our survey participants. While we attempted to cast a very wide net using the sources described in the Survey Participants section, we cannot be certain that the sample who responded to the survey is representative of the overall population of research software developers in the United States.

The second is the discrepancy among the number of respondents of each type (Researcher, Combination, Developer). Because we did not know how many people in the population would identify with each respondent type, we were not able to use this factor in our recruitment strategy. As a result, the number of respondents from each type differs. This discrepancy could be representative of the overall population or it could also suggest a sampling bias.

Therefore, while we have confidence in the results described above, these results may not be generalizable to the larger population.

## DISCUSSION

We turn now to a discussion of the results of our survey, and the implied answers to our research questions. In each subsection, we restate the original research question, highlight important findings, and contextualize these findings in relation to software sustainability.

### Software engineering practices

**RQ1: What activities do research software developers spend their time on, and how does this impact the perceived quality and long-term accessibility of research software?**

Across a number of questions about software engineering practices, our respondents report the aspects of the software development process that were more difficult than expected were related to people, rather than mastering the use of a tool or technique. Respondents reported they thought testing was important, but our results show only a small percentage of respondents frequently use system testing (27% of respondents) and regression testing (24% of respondents). This result suggests a targeted outreach on best practices in testing, broadly, could be a valuable future direction for research software trainers.

We also asked respondents about how they allocated time to software development tasks. The respondents reported, overall, they spend their time efficiently - allocating as much time as a task requires, but rarely more than they perceive necessary (see Fig. 2).

However, one notable exception regards *debugging*, where both developers and researchers report an imbalance between time they would like to spend compared with the time they actually spend. While we did not ask follow up questions about any specific task, we can interpret this finding as the result of asking about an unpleasant task - debugging is not an ideal use of time, even if it is necessary. However, there is an abundance of high quality and openly accessible tools that help software engineers in debugging tasks. A future research direction is to investigate types of code quality controls, testing, and the use of tools to simplify debugging in research software.

Overall, we observe research software developers do not commonly follow the best software engineering practices. Of the practices we included in our survey (Continuous Integration, Coding Standards, Architecture/Design, Requirements, and Peer Code Review), none were used by more than 54% of the respondents. This result indicates a need for additional work to gather information about how research software developers are using these practices and to disseminate that information to the appropriate communities to increase their usage.

## Software tools

### RQ2: What tools do research software developers use and what additional tools are needed to support sustainable development practices?

Our respondents reported sufficient tool support only for the *coding* activity. This result suggests the need for additional tools to support important activities like Requirements, Design, Testing, Debugging, and Maintenance. The availability of useful tools that can fit into developers current workflows can increase the use of these key practices for software quality and sustainability.

## Training

### RQ3: What training is available to research software developers and does this training meet their needs?

Across multiple questions, our findings suggest a need for greater opportunities to access and participate in software training. Previous surveys found less than half (and sometimes much less) of developers reported they had formal training in software development. Our results support these rough estimates, with Developers reporting a slightly higher percentage of both formal and informal software training than researchers. In our sample, including both developers and researchers, approximately half of the people developing research software have received no formal training. Consistent with this number, only about half of the respondents reported sufficient opportunities for training. However, approximately 3/4 indicated they did not have sufficient time for training that was available. Together these results suggest two conclusions: (1) there is a need for more training opportunities (as described above) and (2) developers of research software need more time for training, either by prioritizing in their own schedule or by being given it from their employers.

## Funding

### RQ4:

- **RQ4a: What is the available institutional support for research software development?**
- **RQ4b: What sources of institutional funding are available to research software developers?**

We provided a motivation for this research question by demonstrating, across a variety of previous surveys and published reports, there has not been sufficient funding dedicated to the development and maintenance of research software. The survey results support this

assertion with just under half of the respondents who develop software as part of their research report including costs for developing software into research funding proposals, with even less including costs for reusing or maintaining research software. A limitation of our study is that we do not ask respondents why they choose not to include these costs. We could interpret this result as a belief that such items would not be appropriate for a budget or would not result in a competitive funding application. Future should investigate (1) how and why software research funding is allocated, (2) how research software is budgeted in preparing research grant proposals, and (3) what deters researchers from requesting funding for software development, maintenance, or reuse.

From the perspective of software sustainability, these results are troubling. Without support for maintaining and sustaining research software, at least some of the initial investments made in software are lost over time.

## Career paths

### RQ5: What factors impact career advancement and hiring in research software?

Our results show a diversity of people and their titles who assume the role of research software developer. However, few respondents were optimistic about their research software contributions positively impacting their career (only 21% of faculty and 42% of developers believed software contributions would be valuable for career advancement). This finding was particularly pronounced for female identifying respondents, with only 16% (n = 32/202) believing software contributions could impact their career advancement.

When asked to evaluate prospective applicants to a research software position many respondents valued potential and scientific domain knowledge (background) as important factors. We optimistically believe this result indicates that while programming knowledge, and experience are important criteria for job applicants, search committees are also keen to find growth minded scientists to fill research software positions. We believe this result could suggest an important line of future work - asking, for example, research software engineering communities to consider more direct and transparent methods for eliciting potential and scientific domain knowledge on job application materials.

Finally, we highlight the factors important to research software developers when evaluating a prospective job for their own career. Respondents reported that they value salary equally with leadership (of software at an institution) and access to software resources (*e.g.*, infrastructure). This suggests that while pay is important, the ability to work in a valued environment with access to both mentorship and high quality computing resources can play an important role in attracting and retaining talented research software professionals.

## Credit

**RQ6:**

 • **RQ6a: What do research software projects require for crediting or attributing software use?**

 • **RQ6b: How are individuals and groups given institutional credit for developing research software?**

As described in the Background section, obtaining credit for research software work is still emerging and is not consistently covered by tenure and promotion evaluations. The survey results are consistent with this trend and show none of the traditional methods for extending credit to a research contribution are followed for research software. Figure 15 makes this point clear - respondents most frequently mention software by name but less frequently cite software papers or provide links to the software. According to these results, research software projects require better guidance and infrastructure support for accurately crediting software used in research, both at the individual and institutional level.

We also highlight the relationship between credit and career advancement. In the previous section (Career Paths) we asked respondents how often they were consulted about or asked to contribute to existing software projects at their institution. Among the Developer respondents, 72% were consulted about developing and maintaining software. However, unless this consultation, expertise, and labor is rewarded within a formal academic system of credit, this work remains invisible to tenure and promotion committees. Such invisible labor is typical within information technology professions, but we argue that improving this formal credit system is critical to improving research software sustainability.

Scholarly communications and research software engineers have been active in promoting new ways to facilitate publishing, citing, using persistent identifiers, and establishing authorship guidelines for research software. This effort includes work in software citation aimed at changing publication practices (*Katz et al., 2021*), in software repositories (*Smith, 2021*), and a proposed definition for FAIR software to add software to funder requirements for FAIR research outputs (*Chue Hong et al., 2021*).

## Diversity

**RQ7: How do current Research Software Projects document diversity statements and what support is needed to further diversity initiatives?**

Previous studies of research software communities have not focused specifically on diversity statements, DEI initiatives, or related documentation (*e.g.*, code of conduct documents). Our results showed about 2/3 of the respondents thought their organizations promoted a "culture of inclusion" with respect to research software activities. Conversely, only about 1/3 of respondents thought their organization did an above average job of recruiting, retaining, or meaningfully including diverse groups (see Fig. 18).

We also asked participants whether their main software project (where they spent most time) had, or planned to develop, a diversity and inclusion statement. Over half of

respondents indicated that their projects do not have a diversity/inclusion statement or a code of conduct and have no plans to create one.

While these numbers paint a grim picture, we also believe that there is additional research necessary to clarify the types of diversity, equity, and inclusion work, including formal and informal initiatives, needed in research software development. This research would provide needed clarity on the training, mentoring, and overall state of diversity and inclusion initiatives in research software. Further, research needs to compare these practices with broader software and research communities seeking to understand how, for example, research software compares to open-source software.

## ACKNOWLEDGEMENTS

We thank the survey participants for their time.

### Funding
This work was supported by a grant from the United States National Science Foundation (1743188) "SI2-S2I2 Conceptualization: Conceptualizing a US Research Software Sustainability Institute (URSSI)." The funders had no role in study design, data collection and analysis, decision to publish, or preparation of the manuscript.

### Grant Disclosures
The following grant information was disclosed by the authors:
The United States National Science Foundation: 1743188.

### Competing Interests
Daniel S. Katz and Sandra Gesing are Academic Editors for PeerJ CS.

### Author Contributions
- Jeffrey C. Carver conceived and designed the experiments, performed the experiments, analyzed the data, performed the computation work, prepared figures and/or tables, authored or reviewed drafts of the paper, and approved the final draft.
- Nic Weber conceived and designed the experiments, performed the experiments, analyzed the data, authored or reviewed drafts of the paper, and approved the final draft.
- Karthik Ram conceived and designed the experiments, performed the computation work, prepared figures and/or tables, authored or reviewed drafts of the paper, and approved the final draft.
- Sandra Gesing conceived and designed the experiments, performed the experiments, authored or reviewed drafts of the paper, and approved the final draft.
- Daniel S. Katz conceived and designed the experiments, authored or reviewed drafts of the paper, and approved the final draft.

## Ethics

The following information was supplied relating to ethical approvals (i.e., approving body and any reference numbers):

The University of Notre Dame granted ethical approval to carry out the study (18-08-4802) (Sandra Gesing was at Notre Dame at the time of the survey).

## Data Availability

URSSI Survey Raw Data and Survey Questions are available at Zenodo: Jeffrey Carver, Sandra Gesing, Daniel S. Katz, Karthik Ram, & Nic Weber. (2021). URSSI Community Survey 2018 Raw Data. Zenodo. https://doi.org/10.5281/zenodo.6338766.

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
