# Peer review of "A survey of the state of the practice for research software in the United States"

_PeerJ Computer Science, doi:10.7717/peerj-cs.963_

## Round 0.1 · original submission · Major Revisions

We have received detailed reports for this manuscript. I recommend a major revision before further consideration. Please provide a detailed response letter together with a change tracked revision.

·

Basic reporting

The text is well written, but contains a few syntax and grammar errors (lines 169, 178, 510, 661). In line 316 the sentence structure seems to be off for me. While I (as a non native speaker) cannot determine if it is wrong, it could be improved for better understanding.

The paper is very well structured and figures and tables are included in appropriate places to ease the perceiving of the data, but unfortunately the figures have some drawbacks.

Figure 1 is missing an explanation for the two sets of columns. Also, it is rather unreadable due to the subfigures being too small. Removing at least the vertical dotted lines might further increase readability.

Nearly all figures show absolutes. Especially with the difference regarding the absolute number of answers, I would prefer a presentation as relatives (percentages) to allow a better comparison of the answers in figures itself (i.e. figure 5 and 6) and between figures. My suggestion is to state the absolutes in the figure caption or the x-axis captions since it is an important information. In the text the authors already use mainly the percentages, referring to the absolutes in brackets.

Experimental design

In line 388 it is stated that the data was sanitized, but not explained in what way. A short description of this process would be helpful.

Validity of the findings

The results are US and UK centric, which is due to the nature of the survey distribution and focus and acknowledged by the authors. While I see no issue with this and most findings might be applicable for other regions, I would recommend to mention it earlier in the paper, perhaps even the abstract.

·

Basic reporting

The basic reporting is done well. I indicated on the attached pdf a few spots where the sentences are phrased awkwardly, or where there are some typos, but for the most part the English is quite good.

The literature review is particularly well done.

The figures were nicely done, except for a few points that I highlighted on the marked up pdf.

The paper is self-contained. The results are given for the research questions.

The section with the title Discussion should probably have the section title changed to Conclusions, since the Discussion is really what came in the previous section. Also, there isn't currently a Conclusions section.

Experimental design

The research questions are well defined, except for one where the wording is awkward. The one with the awkward wording is highlighted in the marked up version.

The investigation is well performed and the ethical aspect is explained. It is also nice to see that the full questions and data are released as open-source. It is good to see authors practicing what they preach (or eating their own dog food). :-)

Enough information is given to replicate the experimental design.

Validity of the findings

For the most part the findings seem reasonable. However, there are a few questionable points, as highlighted in the marked up version. A few specific comments include:

1. A bigger deal seems to be made of the differences in Figure 1 than the data justifies. The ideal time and the spent time look similar in all cases. For the cases where the paper claims the difference is noteworthy, it would be good to quantify the relative difference between the two measures.

2. A section on threats to validity is missing. One threat to validity is the respondents not knowing the meaning of the terminology, such as regression testing, requirements, software carpentry, maintaining and sustaining, design, .... Another threat is respondents self-reporting on their activities, such as what documentation they write.

3. There is no section for future work. For instance, the respondents self-report on what documentation they write, but this documentation could be measured directly by mining existing software repositories. Also, extended interviews could be held with some researchers/developers to get more qualitative data.

4. At line 643 it is stated that the most difficult aspects of software development are not technical. However, this is misinterpreting the study question. The respondents weren't asked what they found most difficult, but what aspects of software development are more difficult than they should be. The respondents might feel like the technical aspects are difficult, but not more difficult than they should be.

Additional comments

The paper shows that domain data was collected, but this data does not appear to be used. Did the responses differ between the different domains? If this was not investigated, possibly because of insufficient data, this should be explicitly stated.

Reviewer 3 ·

Basic reporting

The paper is very well-written. The language used throughout is very clear and unambiguous and the research is very well motivated. The introduction is really great and motivates the problem well. The tables and figures are also clear and easy to understand. The background is extensive, however the section is a little unorganized. I appreciate the list of prior surveys and Table I is very helpful, however it would be useful to have more specific details about how these prior studies differ from the current work. Additionally, the research questions (more on this later) do not seem to fit in this section. They have some background from the prior work, but are not well motivated for the overall work. There should be more background included to motivate each RQ and explain why it's included or they should be placed in a separate section (i.e. with the Methods). Overall this research is exciting and provides a valuable contribution to the field.

Experimental design

The research is within the scope of the journal. The investigation and methods are rigorous and meet a technical and ethical standard. The main limitation are the research questions. There is a wide breadth between the RQs and, while the researchers show how these topics fit into the previous surveys, in my opinion the authors fail to motivate why these topics are important for this study and show how each RQ relates to each other and contributes to the overall goal of the work. Each RQ on its own seems very meaningful and relevant, however it is unclear how these connect to each other. To improve upon this, I would suggest adding to each RQ to explain why it is necessary, how it was selected, and how it fits into the current work. I think also combining the RQs with the specific parts of the survey mentioned in the Methods section would also be helpful. Additionally, the RQs lack background for some of specifics studied, especially for the SE-related questions. For example, requirements, design, and testing as software engineering processes, however other potential processes considered part of SE such as implementation, deployment, maintenance, etc. are missing in RQ1. Also the *best* practices (continuous integration, coding standards, arch/design, requirements, peer code review) have some missing options (i.e. pair programming, static analysis tools,) and would be interesting to know the details on how these were selected.

Validity of the findings

The work is novel and impactful for improving how research software is developed. The findings are valid and sound based on the methods and RQs. It would be beneficial to see statistical tests used to further analyze the results and provide their significance. The organization of the results could also be improved by providing a clear answer (i.e. highlighted) to each of the research questions based on the findings. The paper is also missing a threats to validity section to explain limitations of the work (i.e. the huge discrepancy between the respondent types in the results) and how the researchers mitigated these threats. In the conclusion, I would prefer more discussion about implications of these problems and solutions for researchers and developers to improve the state of research software development for each RQ topic, especially for improving the software engineering practices. Additionally, it would be interesting to note future opportunities for this work, such as mining GitHub repos to programmatically determine the tools and practices used for research software. Overall this is a very interesting research paper with implications for improving research software, development practices, and research overall. Looking forward to seeing what's next!

·

Basic reporting

The text follows established, scientific form and quality. I would even go as far and say that especially the text up until the "Analysis" section is written well above average quality.

However, starting with the "Analysis" section, it feels like the publication was written by entirely different authors. This (long) section contains mostly tables, figures, and just a little text to describe the former.
The tables and figures could be improved in several places (see detailed notes). Especially for the figures I would suggest to rethink again what they are supposed to show and whether their respective current format is really the best way to display that. Detailed comments can be found later in the review.

I am not sure if this is an artifact of the review-pdf compilation, but almost all citations miss their dois and some citations are not meaningful without either a doi or an URL. Details can be found later in the review.

Up until line 247 (~1/3 of the paper), the text is written in a country-neutral way (including title and abstract), leaving the reader under the impression that the whole RSE landscape is under review. However, as the authors mention themselves: "results differ across the world". Thus, I strongly suggest to mention either in the title or at at least in the abstract the limitation that this survey primarily covers only the USA.

Experimental design

The raw answer data is provided, along with a (pdf) description of the questions. Something I miss is documentation of the mentioned partial dependencies of questions on answers of previous questions. This unnecessarily complicates a repetition or comparison with other surveys. In addition to providing these inter-dependencies in the supplementary material, it would be beneficial to the reader of the pdf-part of the publication if those could be highlighted or marked in some way.

What I miss most is the source code of the scripts used to generate the results from the raw data, including the same for generating the figures. While it is not difficult to do this with the provided csv data from scratch, the vast number of reported numbers and plots would make scripts to generate all of those valuable, and in case of discrepancies between own and reported results the reader would be left without clear way to see where the differences are, in short: they would be very valuable to replicate the results from the data. This is especially disappointing given the connection of this issue with the topic of the paper and the background of the authors. It is entirely possible that I missed those scripts, but I could not find them either in the files submitted to the journal, nor in the data published on Zenodo.

Validity of the findings

All raw data used in the report has been provided. Repetition would be hindered by the lack of dependency information and scripts for the analysis of the raw data, as mentioned in review parts 1 and 2.

Conclusions are overall well stated, with only few exceptions that are mentioned in the detailed review.

Additional comments

While reading the article, I collected notes of everything I spotted, even small issues like typos and the like. In the following, those notes are given in roughly the order of appearance in the text. This also means that rather more important issues are mixed with rather trivial ones . I hope this order helps to speed up the improvement of this article, because I believe it to be valuable to the scientific community.

- line 80: "hey often focus on small groups or in laboratory settings" → something is off here. Maybe remove the "in"
- line 84: "as briefly described next in the next subsection" → drop the first "next"
- line 145: "Software Saved International Survey" is not the proper name of the survey, but rather generated from the github project name. It is commonly known as and should be better named "SSI international RSE survey".
- line 169: missing parentheses around Philippe et al. (2019)
- line 169: double-parentheses around Pinto et al., 2018; Hannay et al., 2009
- line 178: there is a space missing in between "integration(Shahin"
- line 199: "(Nangia and Katz, 2017)": the same publication was already cited within the same sentence context. I suggest to drop the second reference.
- line 201: The sentence "The AlNoamany and Borghi (2018) survey reported similar results" seems to first suggest it reports similar results as the previous sentence, which mentions a gender gap. However, reading further, the reference "similar results" seems to refer to the gender-neutral percentage of training which was discussed further up. While not technically wrong, I suggest to reword this a little to make it easier to understand the reference.
- line 207: This sentence contradicts the sentence before directly, in which it is stated that only a minority of researchers (~20%) found "formal training to be important or very important". If the authors intended the meaning that this result may be caused by a lack of said training ("They don't put importance into something because they don't know it"), this should be made clearer.
- line 276: "Therefore" seems to suggest to refer to the previous sentence: "The results showed that...", but this does not make sense here. It would make more sense to refer to the much previously mentioned "most of the previous surveys did not address the topic of credit": please rephrase to make this clearer.
- line 279: "asking individuals who, ..., directly have important information" → "asking individuals who, ..., directly have the relevant information"
- line 290: "conferences papers" → "conference papers"
- line 292: "(e.g. hiring" → "(e.g., hiring" (comma)
- line 306: "as is occurs" → "as occurs"
- line 330: To the reader, it is not immediately clear that the statement about politeness is also a result of the earlier-quoted work by Ortu et al. (which it is). A solution could be "However, they also show that this demographic ..."
- line 366: "topics" → "topic"
- line 401: given that developers are also focus of later plots it would be good to also mention their number here, even if it can be calculated at this point.
- table 2: numbers in "Administrative & business studies" seem to be mixed-up ("10 0" in one cell, nothing in the last cell, and indeed there seems to be an "&" missing in the source file)
- table 2: minor: right-aligning the numbers would look a little nicer
- figure 1: It is not clear which group is which in each sub-plot (left/right vs. combination/developers?). It is also not mentioned in the lines that describe the plot: lines 435-438. In general, the figure also suffers from the small numbers displayed in each already small sub-plot. I wonder whether it would be better to switch axes and put all 10 sub-plots as rows into one plot which can then run the entire text width. Also (but hard to see and judge as it is right now), it might be worth noting that at least the left group (combination?) wishes for more training time than they actually use.
- caption figure 1: Why capitalize each word?
- line 442: "The only one that is technical is testing." It might be worth noting that this is only a particular high result for the 'combination' group, but not the 'developers' group. On the other hand, for those, "requirements engineering" is comparatively high, while for the 'combination' group it is not.
- lines 444-446: if possible during type-setting the paper for paper-format, it would be nice to use the space of these two lines for the figures and put the content of the lines on the next page.
- figure 2: It would be better to plot the two differently-colored bars side-by-side (like in figure 3). This way, the two groups can be more easily compared between bars of the same color. If, on the other hand, the differently-colored bars are meant to be "behind each other", with the red always being "in front of" the yellow bar, this should be indicated. Even better would be a percentage-based graph, which would also make it possible to compare the two respondent groups with each other, given their different size.
Also, with ~50 being the highest number seen here for groups of a size a factor of 10 larger, I wonder whether this was a multiple-choice-question or a single choice question. The difference matters, as with a multiple-choice question a vast majority effectively answered for every aspect that it is not more difficult than it needs to be, while for a single-choice question only the "most annoying" aspect could be selected, effectively producing a potentially drastically different plot than when asking the multiple-choice version.
- line 444: "Focusing on the one technical aspect that respondents perceived to be more difficult than it should be": this would imply that "testing" would be the only technical aspect where respondents answered that they would be more difficult than they would be. However, figure 2 shows a lot of higher-than-zero answers for also other technical topics. "Testing" is the one technical aspect that was selected most often when it comes to being more difficult than it should be, but not the only one.
- figure 3: It is not clear which group forms the basis for the plot (combination or developer or the sum of both). The numbers in the text give indication that it is likely either combination or combination + developer, but it is not explicitly mentioned.
- line 450: Within the first sentence, it is again not clear which group was takes as basis for those numbers. Given that this was already an issue for more than one question, it might be a good idea to add a general indicator in the paper for each question that marks which of the groups received that particular question and that numbers without group statement always mean "everyone who received that particular question".
- line 467: "we see a different story": not necessarily. The given list of topics to document might not contain, or appear not to contain, the type of documentation applied when documenting code in-line. For instance, a comment on a particular loop structure might not appear to be within "software design" it respondents correspond that answer with the general design of the overall software package.
- figure 4: The text about the figure concentrates around percentages (e.g.: "Less than 30% of the respondents reported"). Therefore, changing the figure to use percentages would help the reader. Although the figure itself will show large differences, the y-axis will. Also, since the text combines 'extremely supported" and "very supported", it would be helpful to visualize those separately from the others, either by using similar colors (and dissimilar to the others), or to add a small black bar between these two and the rest. In addition, if the current order of the topics does not have special meaning, it might be good to sort them by, e.g., the percentage of combined "extremely supported" and "very supported".
- line 484: From the text it is not clear which group received that question, i.e., which concrete dependency was connected to this question.
- line 488: The "However" might not be warranted. It does imply that an overlap of respondents who answered "use git most of the time" and "use copy/zip most of the time" would be noteworthy and possibly troublesome. An alternative explanation might be that those with overlap indeed use both, to achieve an even higher level of backup than one single practice alone.
- line 491: "The lack of use of standard version control methods": The authors state earlier, that "83/87 of respondents answered to use version control. I cannot see a large lack of use of standard version control methods given those numbers.
- line 500: It would be interesting to look at the reason for the (small) difference. I could imaging that different gender ratios in specific disciplines and different availability of training in those disciplines could have an effect of comparable size. While the text does not explicitly state a causation, it might be good to explicitly state that this is indeed by itself no indication for causation.
- line 510: double "half half": the first likely should be a "than"
- lines 511/514: like at line 500 it would be interesting (and should be within the data collected) to see whether this could be a correlation with something else, like differences in availability of training and gender ratio differences in different disciplines.
- figure 5: plotting percentages would allow comparisons between the different groups, while plotting absolute numbers do not really add benefit.
- figure 8: percentages would be a better choice to show the data.
- figure 9: same as for figure 8, but this is even more important here, where the current plot does not really enable a good comparison between the two groups shown here.
- figure 10: also here, a percentage plot would be more useful than absolute numbers
- line 563: The text states "Figure 10 shows, the respondents saw little chance for career advancement for those whose primary job is software development". However, when looking at figure 10, the group of "developers" show a far better opportunity for advancement (50/50 no/yes) when compared to the other two groups (far lower numbers of "yes" compared to "no"). Also, it is interesting to see for the group of researchers only, the group of "don't know" is the largest overall.
- line 565: with so big differences between groups in figure 10, I wonder how relevant the statement concerning gender is really for the gender topic and how much could result from different gender ratios in the groups plotted here.
- figure 11: percentages would be a little more useful. Luckily, the total numbers are somewhat similar here.
- figures 12&13: percentages would be really important here, as the absolute numbers for developers and the combination group are so different that gets really hard to see the composition of the numbers for developers.
- caption of figure 12: "sfaff" → "staff"
- line 597: I would argue that availability of resources is about as important then the other three mentioned here, for the combination group reaching 30 for 'extremely' and 'very' important combined, and even if hard to see, this seems to be even more important for the group of developers. Thus, I feel it should be mentioned as important as well.
- caption of figure 15: "current" → "currently"
- caption of figure 16: "Does" → "do", "casess" → "cases"
- figure 18: percentages would help comparing the different groups with each other.
- line 634: "This answer again seems at odds"...: Not necessarily. Assuming most of these projects are within scientific institutions, those more likely do have one of both of these, so while the project might not have any, it would be covered by the institutional one (but the question only asked about the project specifically). Also, there are different views of the effectiveness of these measures, which might lead projects to adopt different practices to further inclusion.
- figures 19, 20 and 21 are for some reason displayed differently than most of the others before: here, the groups use different colors and the answers are displayed as different bars, while it is the other way around for the figures beforehand. It would be easier for the reader if this would be done more consistently. Besides consistency, using percentages instead of absolute numbers (and swapping variables as mentioned above) would the reader judge differences between different groups a lot easier.
- line 653: I would argue that any time spent on debugging is too much, as ideally there would not be any need for debugging. In this sense, it is not surprising that a lot of people responded that they spend more time for debugging than they ideally would (because that would be no time at all).
- line 661: I suggest to add an "of" inside of "the development *of* research software"
- line 721 should likely be cited as specified on F1000: https://f1000research.com/articles/9-295
- line 724: Is there a DOI or an URL available for reference?
- line 725: There seem to be two (different) dashes in between the page numbers.
- line 726: Likely, at least the word "interdependencies" should start with a capital "I"
- line 728: URSSI should be all capital; also a link to the data or a doi would be nice.
- line 744: UK should be all capital
- line 793: "investigating" likely should start capitalized
- line 797-799: The doi to the data is missing: 10.5281/zenodo.2585783. Without the doi, the citation is hard to use: a reader would not even know that the name refers to a github organization without utilizing a search engine. Also, a link to an analysis of the relevant data would be interesting to readers: https://www.software.ac.uk/blog/2018-03-12-what-do-we-know-about-rses-results-our-international-surveys
- line 800: "an" should likely start capitalized
- line 811: That citation should at least include the link to the blog entry. As it is, one would have to rely on a search engine to try to find what is meant.
- line 819: There seem to be two dashes: "CIDCR1––CIDCR7"
- line 820: An URL would be useful.
- line 828: There are two dashes in between the page numbers.
- This may be an issue with the reviewer version of the PDF, but clickable links, both internal (footnotes, citations), as well as external (dois/URLs) should work.
- p.2, footnote 6: URL uses different font than other URLs (and probably should be an ssl-URL too)
- p.6, footnotes 7011: URLs use a different font than URLs in footnotes 3-5. Given the length and subsequent unreadable line-break in the URL of footnote 11, I suggest to try the (smaller) font of the URLs 3-5 here.
- Aside from line 766, all references seem to be missing their dois.
- All figures seem to have been provided in png format. However, being a raster-format, this might produce low-quality results for various zoom-levels. Please consider using scalable formats instead for plots like the ones in this publication.

---

## Round 0.2 · Minor Revisions

A further revision is needed. Thanks.

·

Basic reporting

The edits to the basic reporting show attention to detail and respect for the reviewer comments. The highlighted concerns seem to have been addressed. However, some of the edits could be improved as follows:

1. Figure 1 is difficult to read. The font size is too small for the titles of the subplots.

2. Figures 1 and 7 show dots that are confusing. The text does not say what the dots represent. They could be data points for individual responses, but then they don't show the multiplicity when more than one respondent has the same answer. Also, there are cases (like NIH for Developers and DoE for Developers in Figure 7) where there are no "data points" below the mean. This makes me think that I am misinterpreting the dots. They should either be explained, or, if they don't convey any useful information, removed.

3. Although the response to reviewers stated that a Conclusions section was added, there still isn't a Conclusions section. The authors should add a Conclusions section.

Experimental design

The edits to the experimental design address the reviewer comments.

Validity of the findings

The edits to this section more accurately reflect the story that is told by the data, and the limitations of the data.

Additional comments

no comment

---

## Round 0.3 · accepted · Accept

The paper can be accepted. Congratulations.

·

Basic reporting

My concerns have all been addressed.

Experimental design

My concerns have all been addressed.

Validity of the findings

My concerns have all been addressed.

Additional comments

My concerns have all been addressed.